# Exploring the expression patterns of palmitoylating and de-palmitoylating enzymes in the mouse brain using the curated RNA-seq database BrainPalmSeq

**Angela R Wild[1], Peter W Hogg[1], Stephane Flibotte[2], Glory G Nasseri[1], Rocio B Hollman[1], Danya Abazari[1], Kurt Haas[1], Shernaz X Bamji[1]***

[1]Department of Cellular and Physiological Sciences, Life Sciences Institute and Djavad Mowafaghian Centre for Brain Health, University of British Columbia, Vancouver, Canada; [2]Life Sciences Institute Bioinformatics Facility, University of British Columbia, Vancouver, Canada

**Abstract** Protein *S*-palmitoylation is a reversible post-translational lipid modification that plays a critical role in neuronal development and plasticity, while dysregulated *S*-palmitoylation underlies a number of severe neurological disorders. Dynamic *S*-palmitoylation is regulated by a large family of ZDHHC palmitoylating enzymes, their accessory proteins, and a small number of known de-palmitoylating enzymes. Here, we curated and analyzed expression data for the proteins that regulate *S*-palmitoylation from publicly available RNAseq datasets, providing a comprehensive overview of their distribution in the mouse nervous system. We developed a web-tool that enables interactive visualization of the expression patterns for these proteins in the nervous system (http://brain-palmseq.med.ubc.ca/), and explored this resource to find region and cell-type specific expression patterns that give insight into the function of palmitoylating and de-palmitoylating enzymes in the brain and neurological disorders. We found coordinated expression of ZDHHC enzymes with their accessory proteins, de-palmitoylating enzymes and other brain-expressed genes that included an enrichment of *S*-palmitoylation substrates. Finally, we utilized ZDHHC expression patterns to predict and validate palmitoylating enzyme-substrate interactions.

**\*For correspondence:**
shernaz.bamji@ubc.ca

**Competing interest:** The authors declare that no competing interests exist.

## Editor's evaluation

This paper will be of broad interest to neuroscientists, providing a rich resource for future research. Using available RNAseq data the authors build an easy-to-work-with web platform which will enable researchers to survey the expression patterns of palmitoylating and de-palmitoylating enzymes and their potential co-expressed substrates within the mouse nervous system. Using this map, the authors test hypotheses about the relationship between these enzymes and neurological diseases and generate hypotheses about enzyme/substrate relationships based on expression correlations.

## Introduction

Protein *S*-palmitoylation is a post-translational lipid modification that mediates dynamic changes in protein stability, function, and membrane localization. *S*-palmitoylation is defined as the reversible formation of a cysteine residue thioester bond with the fatty acid palmitate, and is the most prevalent post-translational lipid modification in the brain. Dynamic changes in *S*-palmitoylation are critical for neuronal development and synaptic plasticity (*Fukata et al., 2013*; *Fukata and Fukata, 2010*; *Globa

*and Bamji, 2017*; *Matt et al., 2019*), oligodendrocyte differentiation and myelination (*Ma et al., 2022*; *Schneider et al., 2005*), and astrocyte proliferation (*Yuan et al., 2021*). Furthermore, numerous neurological and psychiatric diseases have now been attributed to mutations in the genes encoding palmitoylating and de-palmitoylating enzymes, including schizophrenia, intellectual disability and CLN1 disease (*Mukai et al., 2004*; *Nita et al., 2016*; *Raymond et al., 2007*), underscoring the importance of proper regulation of *S*-palmitoylation for normal brain function.

*S*-Palmitoylation is catalyzed by a family of ZDHHC enzymes that share a consensus 'Asp-His-His-Cys' catalytic domain. These enzymes are structurally heterogeneous multi-pass transmembrane proteins that localize to a variety of subcellular compartments, including the Golgi apparatus, endoplasmic reticulum (ER), recycling endosomes and the plasma membrane (*Globa and Bamji, 2017*). The ZDHHC enzymes are known to associate with accessory proteins that regulate their stability, activity, and trafficking (*Salaun et al., 2020*). Several de-palmitoylating enzymes have also been identified that are divided into three classes: the acyl-protein thioesterases that shuttle between the Golgi and cytosol (APTs; *Vartak et al., 2014*), the predominantly lysosomal palmitoyl-protein thioesterases (PPTs; *Koster and Yoshii, 2019*) and the more recently discovered α/β hydrolase domain-containing proteins (ABHDs; *Lin and Conibear, 2015*). Unlike other post-translational modifications, palmitoylation lacks a consensus substrate amino sequence. To date, there is no unifying theory to explain how substrate recognition is achieved, with contrasting reports of substrate interactions being both promiscuous and specific (*Malgapo and Linder, 2021*). Currently, these interactions are thought to be governed by the subcellular targeting of ZDHHCs enzymes and the presence of protein-protein interacting motifs within the ZDHHC N- and C-termini, which are highly diverse among the ZDHHC enzymes (*Rana et al., 2019*). Differential gene expression can also have a profound influence on protein interactions and may play a role in the coordination of *S*-palmitoylation in the brain. However, a detailed overview and analysis of the precise cellular and regional expression patterns of the palmitoylating and de-palmitoylating enzymes has not yet been described, and as such, little is known about how this expression is coordinated in the nervous system.

Recent advances in single-cell RNA sequencing (scRNAseq) techniques have enabled the classification of neuronal and non-neuronal cell types in unprecedented detail, providing a better understanding of cellular diversity and function in the nervous system, while also providing a means to study the expression patterns of individual genes across an ever-expanding range of brain regions and cellular classifications. Here, we capitalized on the recent surge in RNAseq publications characterizing regional and cellular transcriptomics of the mouse nervous system. We curated and analyzed expression data from a number of publicly available RNAseq mouse datasets to generate a detailed analysis of the expression patterns of the genes associated with *S*-palmitoylation in the mouse brain. Furthermore, we present an interactive web tool that allows user-driven interrogation of the expression patterns of palmitoylating and de-palmitoylating enzymes from numerous collated studies across a variety of brain regions and cell types (http://brainpalmseq.med.ubc.ca/). We demonstrate the utility of this resource by detailing the considerable cell-type and regional heterogeneity in expression patterns of these enzyme families and their accessory proteins, revealing numerous cell-type enrichments and co-expression patterns that allowed us to generate and test hypotheses about palmitoylating enzyme-substrate interactions.

## Results

### BrainPalmSeq: an interactive database to search palmitoylating and depalmitoylating enzyme expression in the mouse brain

The recent development of scRNAseq has revolutionized our understanding of the complex transcriptional diversity of neuronal and non-neuronal cell types in the brain. We found however, there were several barriers to the easy access for much of this data, with no single resource available to evaluate multi-study expression datasets. Data can also be difficult to access when studies are not accompanied by an interactive online web viewer, while datasets that do have a web viewer employ diverse interfaces that are often complex, particularly for large scRNAseq datasets. Furthermore, the differing study-specific analysis pipelines, as well as the variety of data presentation formats in web viewers including heatmaps, bar charts, tables or t-SNE plots can make datasets difficult for non-bioinformaticians to interpret and compare. In order to remove these barriers and provide easy

**Table 1.** RNAseq published datasets curated to create BrainPalmSeq.

| First Author | Year | PMID | Regions | Age | Website | Technique | Sample | Data Accession |
|---|---|---|---|---|---|---|---|---|
| Zeisel | 2018 | 30096314 | CNS/PNS | Juvenile | http://mousebrain.org/ | 10 X Genomics | Single-cell | SRP135960 |
| Saunders | 2018 | 30096299 | Whole brain | Adult (P60-70) | http://dropviz.org/ | Drop-seq | Single-cell | GSE116470 |
| Sjöstedt | 2020 | 32139519 | Whole brain | Adult | https://www.proteinatlas.org/ | RNA-seq | Bulk tissue | |
| Rosenberg | 2018 | 29545511 | Whole brain | Neonatal | n/a | SPLiT-seq | Single-cell | GSE110823 |
| Cembrowski | 2016 | 27113915 | Hippocampus | Adult | http://hipposeq.janelia.org/ | Genetic labeling; RNA-seq | Cell sorted | GSE74985 |
| Zhang | 2014 | 25186741 | Cortex | Adult | http://web.stanford.edu/group/barres_lab/brain_rnaseq.https | PAN; FACS; RNA-seq | Cell sorted | GSE52564 |
| Zeisel | 2015 | 25700174 | Cortex, hippocampus CA1 | Adult | http://linnarssonlab.org/cortex | Fluidigm C1 | Single-cell | GSE60361 |
| Yao | 2021 | 34004146 | Isocortex/ hippocampus | Adult (P50+) | https://celltypes.brain-map.org/rnaseq/mouse_ctx-hip_10x | 10 X Genomics | Single-cell | GSE185862 |
| Kozareva | 2020 | 24259518 | Cerebellum | Adult | https://singlecell.broadinstitute.org/single_cell/study/SCP795 | snSeq; 10 x Chromium V3 | Single-cell | GSE165371 |
| Phillips | 2019 | 31527803 | Thalamic excitatory neurons | Adult | https://thalamoseq.janelia.org/ | NextSeq 550 | Single-cell/ bulk tissue | GSE133911; GSE133912 |
| Chen | 2017 | 28355573 | Hypothalamus | Adult | n/a | Drop-seq | Single-cell | GSE87544 |
| Gocke | 2016 | 27425622 | Striatum | Adult | n/a | Smart-seq2 | Single-cell | GSE82187 |
| O'Leary | 2020 | 32869744 | Amygdala excitatory neurons | Adult | https://scrnaseq.janelia.org/amygdala | scRNAseq | Single-cell | GSE148866 |
| Marques | 2016 | 27284195 | Oligodendrocytes whole brain | Juvenile; adult | http://linnarssonlab.org/oligodendrocytes/ | Fluidigm C1 | Single-cell | GSE75330 |
| Batiuk | 2020 | 32139688 | Astrocytes in cortex/ hippocampus | Adult (P56+) | https://holt-sc.glialab.org/sc/ | Smart-seq2 | Single-cell | GSE114000 |
| Li | 2019 | 30606613 | Microglia in brain | Embryonic, juvenile, adult | https://www.brainrnaseq.org/ | Smart-seq2 | Single-cell | GSE123025 |

access to expression data for the proteins that regulate *S*-palmitoylation in the brain, we created 'BrainPalmSeq', an easy-to-use web platform allowing user-driven interrogation of compiled multi-study expression data at cellular resolution through simple interactive heatmaps that are populated according to user-selected brain regions, cell-types or genes of interest (http://brainpalmseq.med.ubc.ca/).

To create BrainPalmSeq, we first curated three large datasets from whole-brain scRNAseq studies that were acquired through selection-free cell sampling to provide high-resolution expression data covering hundreds of cell types at a variety of developmental ages (*Rosenberg et al., 2018*; *Saunders et al., 2018*; *Zeisel et al., 2018*). As scRNAseq has several caveats including low sensitivity and high frequency of dropout events leading to incomplete detection of expressed genes (*Haque et al., 2017*), we complemented these datasets, where possible, with curation of several bulk and pooled-cell RNAseq studies that used population-level ensemble measurements from whole-brain and region-specific studies. We further included selected studies for the major glial cell types, and data from the most comprehensive neuron-specific study performed to date by the Allen Institute (*Table 1*). Together, the datasets curated in BrainPalmSeq cover all major regions of the mouse nervous system across a variety of regional and cellular resolutions.

Expression data were extracted from selected studies for the 24 mouse ZDHHC genes (*Zdhhc1-Zdhhc25*, while *Zdhhc10* is omitted), as well as the best characterized de-palmitoylating enzymes

(*Ppt1*, *Lypla1*, *Lypla2*, *Abhd4*, *Abhd6*, *Abhd10*, *Abhd12*, *Abhd13*, *Abhd16a*, *Abhd17a*, *Abhd17b* and *Abhd17c*) and ZDHHC accessory proteins (*Golga7*, *Golga7b* and *Selenok*). Where possible, data were processed from the raw transcripts or unique molecular identifier (UMI) counts using the same normalization protocol to allow for more consistent evaluation of differences in gene expression within datasets. We sampled from RNAseq datasets that used a diverse range of sample collection, processing and analysis techniques, allowing for direct visualization of the relative expression patterns of selected genes within datasets. Users can then validate their observations across complimentary whole-brain or region/cell-type-specific datasets included in BrainPalmSeq. Dropdown menus allow for selection of individual ZDHHC genes or brain regions within each dataset, while the hover tool reveals metadata for each cell type, including neurotransmitter designations and marker genes. We provide download links to all expression data including cell type metadata so that users can replot gene expression profiles in their preferred format. To demonstrate the utility of this resource, we performed a detailed exploration of selected datasets from BrainPalmSeq, revealing how expression patterns can give insights into the function of the palmitoylating and de-palmitoylating enzymes in the mouse brain.

## ZDHHC expression in the nervous system shows regional and cell-type-specific patterning

We began by exploring BrainPalmSeq data curated from the 'MouseBrain' dataset, which provides the broadest overview of expression patterns in the nervous system (*Zeisel et al., 2018*). This scRNAseq study sampled multiple dissected regions from the adolescent (mean age ~P25) mouse central and peripheral nervous systems (CNS and PNS, respectively), identifying 265 transcriptomically unique cell-types (referred to herein as metacells) for which we plotted re-normalized ZDHHC expression values, according to the hierarchical cell-type clustering established by the original study (*Figure 1A*). While ZDHHC expression was detected in all regions of the nervous system, expression of the 24 ZDHHC genes was highly variable across metacell types and clusters. We measured the mean ZDHHC expression within each cluster to gain insight into which cell-types in the nervous system have the greatest overall expression of palmitoylating enzymes. The heatmap rows and columns were ranked (sorted by descending averages) to determine which cell-types had the highest expression of ZDHHCs, and which ZDHHCs were most abundantly expressed across cell-types (*Figure 1B*). Mean ZDHHC expression was particularly high in neurons of the PNS, along with cholinergic/monoaminergic and hindbrain neurons of the CNS. Of the non-neuronal metacell clusters, oligodendrocytes had the highest ZDHHC expression, while other glial cell-types appear at the lower end of the ranking (*Figure 1B*). *Zdhhc20* was the most abundantly expressed ZDHHC, with the highest mean expression across all cell-type clusters, followed by *Zdhhc2*, *Zdhhc17*, *Zdhhc3* and *Zdhhc21*, while expression of *Zdhhc11*, *Zdhhc19* and *Zdhhc25* were negligible. We next clustered neuronal metacells of the PNS and CNS according to the neurotransmitter expression combinations, revealing the highest mean ZDHHC expression was observed in neurons that utilized acetylcholine and nitric oxide as co-neurotransmitters, with cholinergic neurons featuring near the top of the list in several neurotransmitter combinations (*Figure 1C*). Monoaminergic neurons utilizing noradrenaline and serotonin also generally ranked high in the list, consistent with the data in *Figure 1B* that ranked cholinergic and monoaminergic neurons as the metacell cluster with the second highest CNS ZDHHC expression overall, indicating a higher propensity for these cell-types to utilize *S*-palmitoylation as a post-translational mechanism to modify cellular signaling. We performed comparative analysis of ZDHHC expression on another large-scale scRNAseq study of the mouse brain that sampled a variety of cortical and subcortical structures of the adult mouse (P60-P70) (*Saunders et al., 2018*; 'DropViz'; *Figure 1—figure supplement 1A*). We found expression patterns and enrichments to be similar across these two independent, large-scale scRNAseq studies, supporting the general trends observed within the MouseBrain dataset.

To gain insight into the potential networks of ZDHHC enzymes that might work together to coordinate *S*-palmitoylation in different cell types we performed co-expression analysis (Spearman correlation) between ZDHHC genes across all 265 metacell types in the MouseBrain dataset (*Figure 1D*). Neuron-enriched *Zdhhc3*, *Zdhhc8*, *Zdhhc17*, and *Zdhhc21* formed the strongest network of co-expression associations, while glial cell-enriched *Zdhhc2*, *Zdhhc9* and *Zdhhc20* formed less robust correlations with other ZDHHCs. Weaker correlations were observed across the 565 cell-types in the DropViz dataset, which may reflect the absence of the PNS neurons and glia in this study (*Figure 1—figure supplement 1D*). Interestingly, strong correlations were not observed for known pairs of palmitoylating

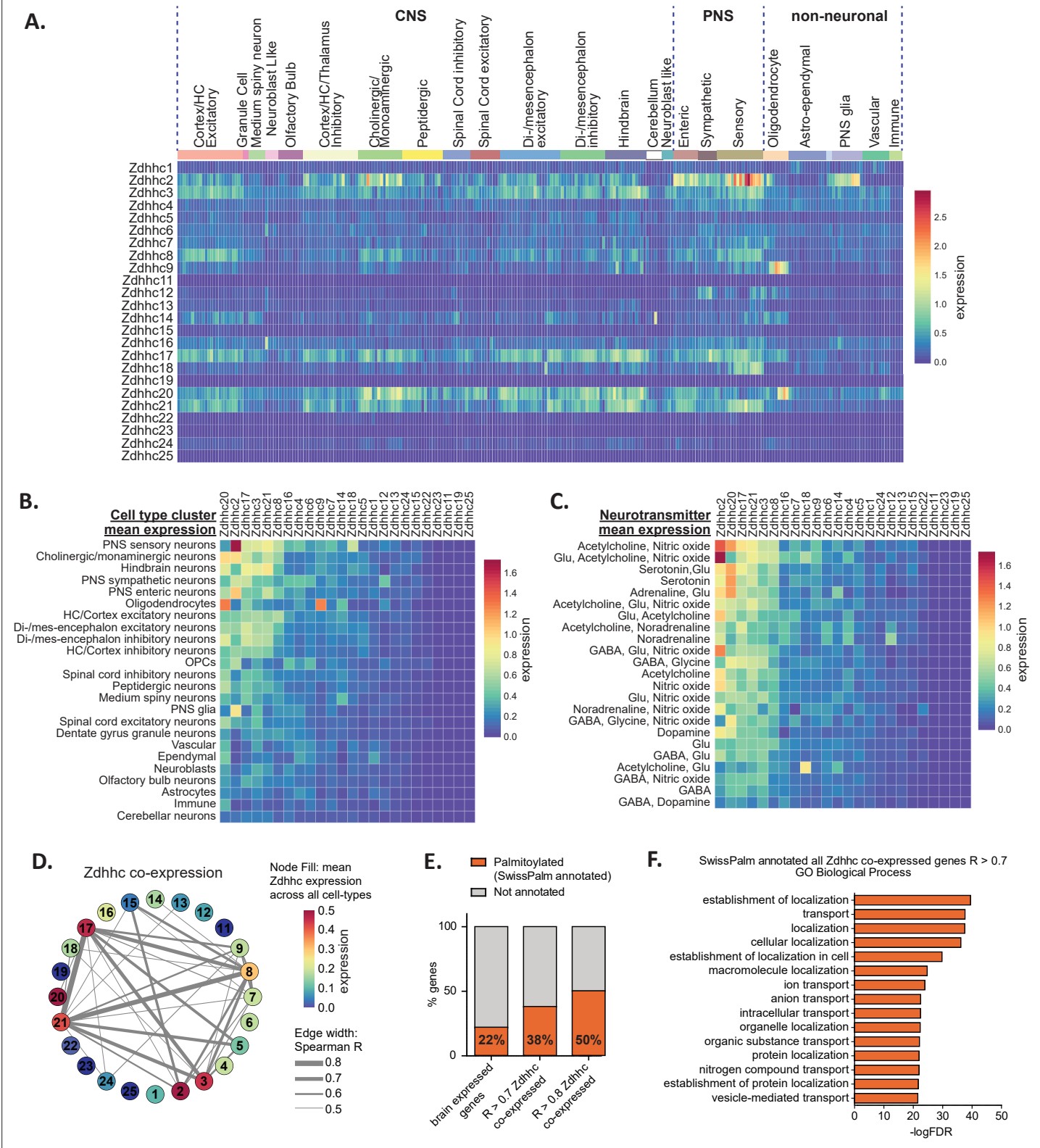

**Figure 1.** Heterogeneous ZDHHC expression in the mouse nervous system. (**A**) Heatmap showing expression for the 24 ZDHHC genes, extracted from scRNAseq study of mouse CNS and PNS (***Zeisel et al., 2018***). Each column represents one of the 265 metacells classified in the study. Metacells are organized along x-axis according to hierarchical clustering designations generated by Zeisel et al. Full metadata for this study available on BrainPalmSeq. (**B**) Heatmap showing mean ZDHHC expression per hierarchical cluster, with columns and rows sorted by descending mean ZDHHC

*Figure 1 continued on next page*

*Figure 1 continued*

expression per row/column. (**C**) Heatmap showing mean ZDHHC expression per neurotransmitter cluster for all PNS and CNS neurons. Columns and rows are sorted as in B. (**D**) Correlation network showing ZDHHC co-expression across all metacells in 'MouseBrain' (Spearman *R*>0.5). Numbers in nodes correspond to ZDHHC number. Node color represents mean expression across all metacells. Edge thickness represents strength of correlation. (**E**) Graph showing proportion of genes from 'MouseBrain' dataset that are co-expressed with one or more ZDHHC and also substrates for *S*-palmitoylation (SwissPalm annotated). 'Brain expressed' n=15,389 protein coding genes expressed in the postnatal mouse brain, curated from the MGI RNAseq studies database. '*R*>0.7 ZDHHC co-expressed' n=914 genes co-expressed with one or more ZDHHC (Spearman *R*>0.7). '*R*>0.8 ZDHHC co-expressed' n=167 genes co-expressed with one or more ZDHHC (Spearman *R*>0.8). Brain expressed vs. *R*>0.7: p<0.001; *R*>0.7 vs *R*>0.8: p<0.01; Fisher's exact test. (**F**) Graph of GO biological process analysis. Gene IDs from the 'MouseBrain' dataset (*Zeisel et al., 2018*) that showed correlated expression with one or more ZDHHC (*R*>0.7) and were also Uniprot reviewed and SwissPalm annotated were used as input. Units for all heatmaps in figure: mean log2(counts per 10,000+1).

The online version of this article includes the following source data and figure supplement(s) for figure 1:

**Source data 1.** ZDHHC expression in the mouse nervous system from 'MouseBrain' dataset (related to *Figure 1A*).

**Source data 2.** ZDHHC cell type averages from 'MouseBrain' dataset (related to *Figure 1B*).

**Source data 3.** ZDHHC neurotransmitter averages from 'MouseBrain' dataset (related to *Figure 1C*).

**Source data 4.** Spearman correlations 'MouseBrain' dataset (related to *Figure 1D*).

**Source data 5.** Spearman correlations of *S*-palmitoylation associated genes vs all other genes in 'MouseBrain' dataset (*R*>0.7) (related to *Figure 1E*).

**Source data 6.** Panther analysis of palmitoylation substrates (SwissPalm annotated) co-expressed with any Zdhhc (*R*>0.7) (related to *Figure 1F*).

**Figure supplement 1.** Heterogeneous ZDHHC expression in the mouse brain.

**Figure supplement 1—source data 1.** Mean cell type ZDHHC expression in the mouse brain system from 'DropViz' dataset (related to *Figure 1—figure supplement 1A*).

**Figure supplement 1—source data 2.** DropViz Zdhhc expression averaged by neurotransmitter (related to *Figure 1—figure supplement 1C*).

**Figure supplement 1—source data 3.** Spearman correlation between Zdhhc genes across 565 cell types in 'DropViz' dataset (related to *Figure 1—figure supplement 1D*).

**Figure supplement 1—source data 4.** Spearman correlations of *S*-palmitoylation associated genes vs all other genes in 'DropViz' dataset (*R*>0.7) (related to *Figure 1—figure supplement 1E*).

**Figure supplement 2.** Correlation of Zdhhc enzymes that are known to be involved in palmitoylation cascades.

enzymes that have been established to be functionally linked, including *Zdhhc20/Zdhhc5* (*Plain et al., 2020*) and *Zdhhc16/Zdhhc6* (*Abrami et al., 2017*). In the case of *Zdhhc20/Zdhhc5*, this could be due to the widespread expression of *Zdhhc20* throughout both neuronal and glial cell types in the nervous system, while *Zdhhc5* is predominantly expressed in neuronal cells. We plotted the expression values for each of the known pairs across 265 identified cell types from 'MouseBrain' dataset (*Figure 4—figure supplement 2*; data available for download on website). Co-expression was variable across cell types, indicating that the relationship between these pairs of enzymes may be cell-type specific.

In order to create a list of potential substrates for the ZDHHCs in the mouse nervous system, we expanded our co-expression analysis to include all expressed genes from the 'MouseBrain' dataset that had significant correlation (*R*>0.7) with one or more ZDHHC. We identified 914 genes with expression patterns that were significantly correlated with ZDHHCs. This list was cross-referenced with the mouse SwissPalm database of *S*-palmitoylated substrates identified in at least one palmitoyl-proteome or experimentally validated (SwissPalm annotated; *Blanc et al., 2015*; *Blanc et al., 2019*). We found that genes that showed correlated expression with a ZDHHC were significantly enriched with *S*-palmitoylation substrates, indicating that ZDHHCs are more likely to be co-expressed with their *S*-palmitoylation substrates in the brain (*Figure 1E*). Co-expression analysis of the 'DropViz' dataset revealed a similar enrichment of *S*-palmitoylation substrates that were co-expressed with ZDHHCs (*Figure 1—figure supplement 1E*), supporting the notion of ZDHHC enzyme-substrate co-expression. PANTHER GO analysis of the ZDHHC co-expressed *S*-palmitoylation substrates curated from 'MouseBrain' revealed several significant enrichments in GO terms for biological processes related to protein localization and transport (*Figure 1F*). These findings are consistent with the known role of *S*-palmitoylation in regulating protein localization and signaling complexes at cellular membranes.

## Heterogeneity in ZDHHC expression within excitatory neurons of the dorsal hippocampus

The hippocampus is a heavily studied brain region that is critical for learning and memory (***Bird and Burgess, 2008***). A recent pooled-cell RNAseq study of excitatory neurons in the hippocampus revealed extensive regional variability in gene expression profiles of the hippocampal tri-synaptic loop (hipposeq.janelia.org; ***Cembrowski et al., 2016***). In order to clearly visualize if ZDHHC expression also varied within these different cell populations, we projected log transformed expression heatmaps generated in BrainPalmSeq for the 'hipposeq' dorsal-ventral excitatory neuron dataset on to anatomical maps of the dorsal hippocampus (***Figure 2A***). We observed considerable heterogeneity in the regional expression patterns of the ZDHHCs in the hippocampus. Hierarchical clustering analysis revealed that the ZDHHCs could be grouped into those that showed similar expression in all regions, those that were dentate gyrus granule cell (DG) enriched, DG depleted or CA1/CA2 enriched (***Figure 2B***). We generated comparative heatmaps for several scRNAseq studies curated in BrainPalmSeq that also quantified the hippocampal excitatory neuron transcriptome and found similar cross-study expression patterns for many of the ZDHHCs (***Figure 2—figure supplement 1A***). Furthermore, in situ hybridization data from the Allen Institute showed a high degree of overlap with the 'hipposeq' derived ZDHHC expression patterns, supporting the replicability of the expression patterns observed in the 'hipposeq' dataset (***Figure 2—figure supplement 1B***).

We next sought to utilize the 'hipposeq' dataset to determine if there might be regional differences in the expression of *S*-palmitoylation substrates in excitatory neurons of the dorsal hippocampus, which may be potential substrates for regionally enriched ZDHHC enzymes. We set out to create a 'projected palmitoylome' for neurons in each sub-region, consisting of genes that are both enriched in each sub-region vs other sub-regions, and are also SwissPalm annotated. We utilized the enrichment analysis tools built in to hipposeq.janelia.org (see Materials and Methods). Neurons in each hippocampal sub-region expressed unique *S*-palmitoylation substrates that were related to highly divergent functions. We found for CA1 excitatory neurons, which have the highest expression of *Zdhhc2*, *Zdhhc17*, *Zdhhc23*, and *Zdhhc9* (***Figure 2C***), the CA1 enriched projected palmitoylome (***Figure 2D***) generated KEGG pathways related to 'Calcium signaling', 'Glutamatergic synapse' and 'Long term potentiation', supporting the known role for *S*-palmitoylation in CA1 hippocampal synaptic plasticity (***Figure 2E***; ***Ji and Skup, 2021***; ***Matt et al., 2019***). The CA1 projected-palmitoylome was composed of around 46% synaptic proteins (SynGO annotated), with SynGO ontologies related to 'synaptic vesicle exocytosis' and 'synapse organization' (***Figure 2F***; ***Koopmans et al., 2019***). In contrast, the projected-palmitoylome of DG granule cells which have the highest expression of *Zdhhc21*, *Zdhhc4*, *Zdhhc24*, and *Zdhhc8* (***Figure 2G and H***) generated KEGG pathways related to 'Ribosome', 'Cholinergic synapse', and 'Parkinson's disease' (***Figure 2I***). The DG projected-palmitoylome was composed of 29% synaptic proteins (SynGO annotated), with SynGO ontologies related to 'protein translation at presynapse' and 'protein translation at postsynapse', revealing a potential role for palmitoylating enzymes in regulating translation in this cell-type that has not yet been studied (***Figure 2J***). Together, we have described patterns of restricted expression of ZDHHC enzymes and *S*-palmitoylation substrates in the dorsal mouse hippocampus, and generated regionally enriched projected-palmitoylomes that provide insight into the role of *S*-palmitoylation in neuronal function in each of these hippocampal sub-regions.

## Neocortical ZDHHC expression is partially segregated across cortical layers and neuronal subclasses

We next examined scRNAseq datasets curated in BrainPalmSeq from the cortex, beginning with a study of the primary somatosensory cortex (SSp; ***Zeisel et al., 2015***). We projected heatmaps of pyramidal excitatory neuron ZDHHC expression generated in BrainPalmSeq onto cortical layer diagrams of SSp, again revealing anatomically heterogeneous excitatory neuron expression patterns for several of the ZDHHC transcripts (***Figure 3A***). Clustering primarily grouped the enzymes according to expression levels, with *Zdhhc21*, *Zdhhc17* and *Zdhhc8* displaying the highest relative expression (***Figure 3B***). *Zdhhc2*, *Zdhhc3*, and *Zdhhc20* expression was also high, with the remainder of the ZDHHCs having moderate-to-low expression. We compared these expression patterns with other datasets curated in BrainPalmSeq (***Figure 3—figure supplement 1A***), which revealed many consistent patterns of expression maintained across several independent studies. For example, multiple studies reported

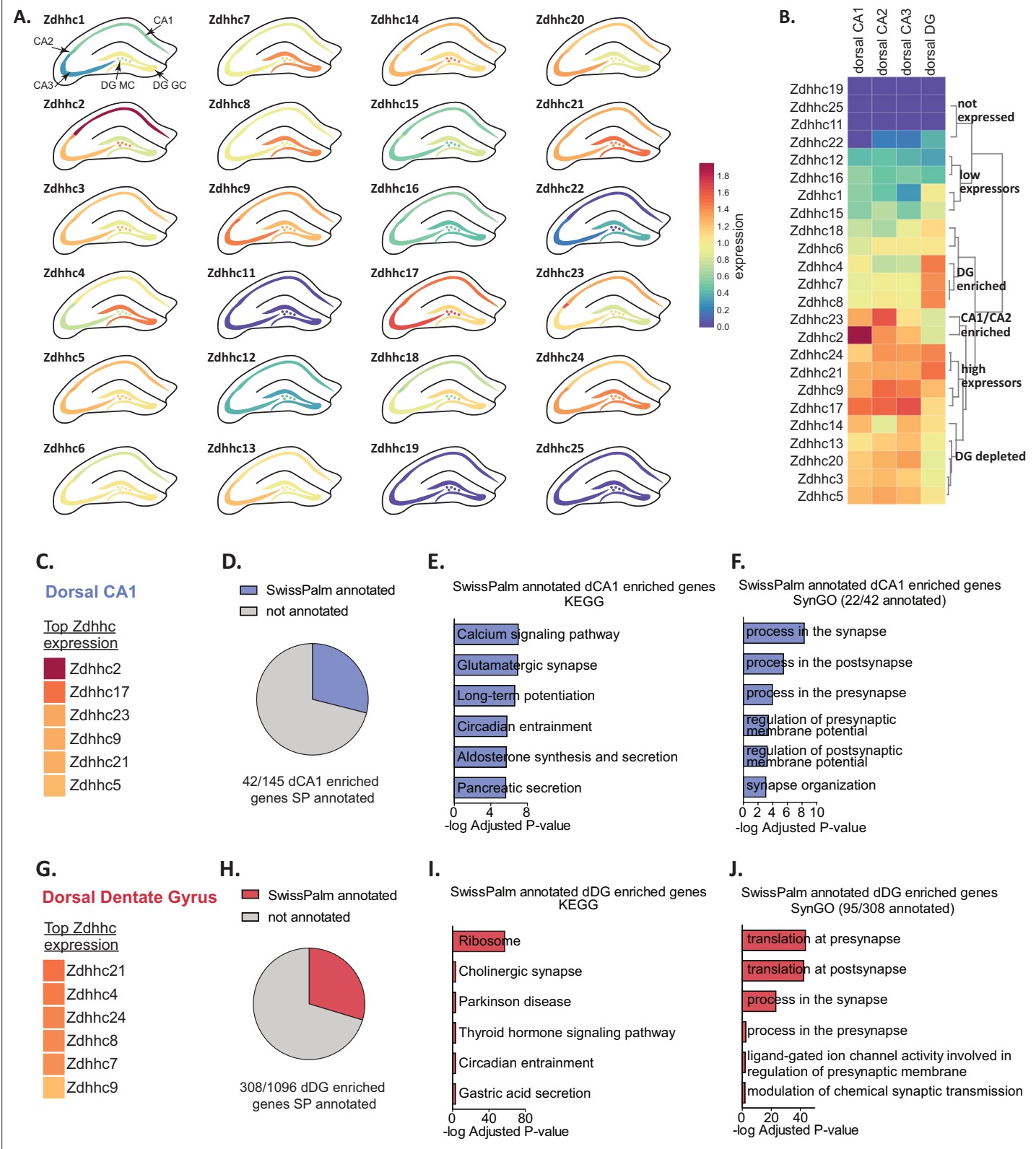

**Figure 2.** Diversity in ZDHHC expression and S-palmitoylation substrate expression in the hippocampus. (**A**) Heatmap of excitatory neuron ZDHHC expression from dorsal hippocampus (original pooled cell RNAseq data from *Cembrowski et al., 2016*) projected onto diagrams of dorsal hippocampus. (**B**) Hierarchical clustering of ZDHHC expression data in A. (**C**) Heatmap showing top 6 ranked expressing ZDHHCs in dorsal CA1 in descending order. (**D**) Pie chart showing proportion genes with enriched expression in dorsal CA1 (dCA1) that are also substrates for palmitoylation

*Figure 2 continued on next page*

*Figure 2 continued*

(SwissPalm annotated). (**E**) KEGG analysis of the dCA1 enriched/SwissPalm annotated genes. (**F**) SynGO analysis of the dCA1 enriched/SwissPalm annotated genes. (**G–J**) As in (**C**)-(**F**) but for the dorsal dentate gyrus (dDG). Heatmap legend in (**A**) applies to all heatmaps (logFPKM +1).

The online version of this article includes the following source data and figure supplement(s) for figure 2:

**Source data 1.** Neuronal expression of Zdhhc genes in dorsal hippocampus from 'HippoSeq' dataset (related to *Figure 2A, B, C and G*).

**Source data 2.** Dorsal CA1 (dCA1) enriched projected palmitoylome (related to *Figure 2D, E and F*).

**Source data 3.** KEGG Analysis of regionally enriched neuron expressed palmitoylation substrates (related to *Figure 2E and I*).

**Source data 4.** SynGO Analysis of regionally enriched neuron expressed palmitoylation substrates (related to *Figure 2F and J*).

**Source data 5.** Dorsal Dentate Gyrus (dDG) enriched projected palmitoylome (related to *Figure 2D, E and F*).

**Figure supplement 1.** Heterogeneous ZDHHC expression in excitatory neurons of the hippocampus.

**Figure supplement 1—source data 1.** Expression of Zdhhcs in hippocampal excitatory neurons across RNAseq studies (related to *Figure 2—figure supplement 1A*).

**Figure supplement 1—source data 2.** In situ hybridization images used from Allan Brain Atlas (related to *Figure 2—figure supplement 1B*).

high expression of *Zdhhc8* in cortical Layer 2/3, enrichment of *Zdhhc2* in Layer 4 and elevated expression of *Zdhhc21* in all layers, particularly in Layer 5. *Zdhhc3* and *Zdhhc20* were also broadly expressed in all cortical layers across studies. Similar patterns were seen in the SSp region from available in situ hybridization studies from Allen Brain Institute (*Figure 3—figure supplement 1B*).

We examined expression patterns of the ZDHHC enzymes from one of the largest neuronal scRNAseq studies from the isocortex performed by the Allen Brain Institute, which identified 236 glutamatergic and 117 GABAergic distinct neuron metacell types (*Yao et al., 2021*). We averaged ZDHHC expression data downloaded from BrainPalmSeq for the major metacell clusters from all regions of the isocortex, according to their anatomical location and/or axon projection and plotted ranked heatmaps (*Figure 3C and D*). *Zdhhc14* was the ZDHHC transcript with the highest expression across glutamatergic neurons, while *Zdhhc2* was one of the highest expressed enzymes in the majority of GABAergic cell-types. Elevated expression of *Zdhhc14* was found in both glutamatergic and GABAergic neurons, which was moderately expressed in other studies of the brain and cortex discussed previously. Glutamatergic neurons in cortical layers 4/5, 5, and 6 featured at the top of the ranking for highest overall expression of ZDHHC enzymes. Neurons within these layers have extensive dendritic branching and long-range axonal projections to the spinal cord, brainstem and midbrain, as well as the ipsilateral cortex, striatum, and thalamus (*Harris and Shepherd, 2015*; *Yao et al., 2021*). GABAergic neurons of the isocortex also showed elevated expression of the common neuronal ZDHHCs including *Zdhhc2*, *Zdhhc3*, *Zdhhc17*, *Zdhhc20,* and *Zdhhc21*. The highest mean ranked expression was observed in the recently categorized Sncg neurons that correspond to Vip[+]/Cck[+] multipolar or basket cells (*Tasic et al., 2018*), and lowest expression was observed in the Vip subclass of interneurons.

Together, our observations reveal that the complex transcriptional diversity of neurons that has recently been revealed by RNA sequencing also includes heterogeneity in the expression of the ZDHHC enzymes that mediate palmitoylation. These expression patterns are likely to influence enzyme-substrate interactions along with the function of *S*-palmitoylation substrates, and thus influence neuronal development, function and synaptic plasticity.

## De-palmitoylating enzyme and ZDHHC accessory protein expression in the nervous system shows regional and cell-type-specific patterning

Dynamic turnover of protein *S*-palmitoylation is mediated by the activity of de-palmitoylating enzymes, which determine the half-life of *S*-palmitoylation on a target protein. Among the first of these enzymes to be discovered were acyl-protein thioesterases 1 and 2 (APT1, APT2; encoded by *Lypla1*, *Lypla2*) and palmitoyl-protein thioesterase 1 (PPT1), which are all members of the serine hydrolase family. However, full characterization of this family of proteins is ongoing, with over 100 members of the serine hydrolase superfamily being identified to date (*Simon and Cravatt, 2010*). Not all these proteins have been found to exhibit depalmitoylating activity, therefore we limited our study to include those that are blocked by palmitoylation inhibitors that mimic the palmitoyl acyl chain, including Palmostatin B and hexadecylfluorophosphonate (*Lin et al., 2017*). These included a number of the α/β hydrolase domain-containing proteins (ABHD4, ABHD6, ABHD10, ABHD12, ABHD13, ABHD16A, ABHD17A,

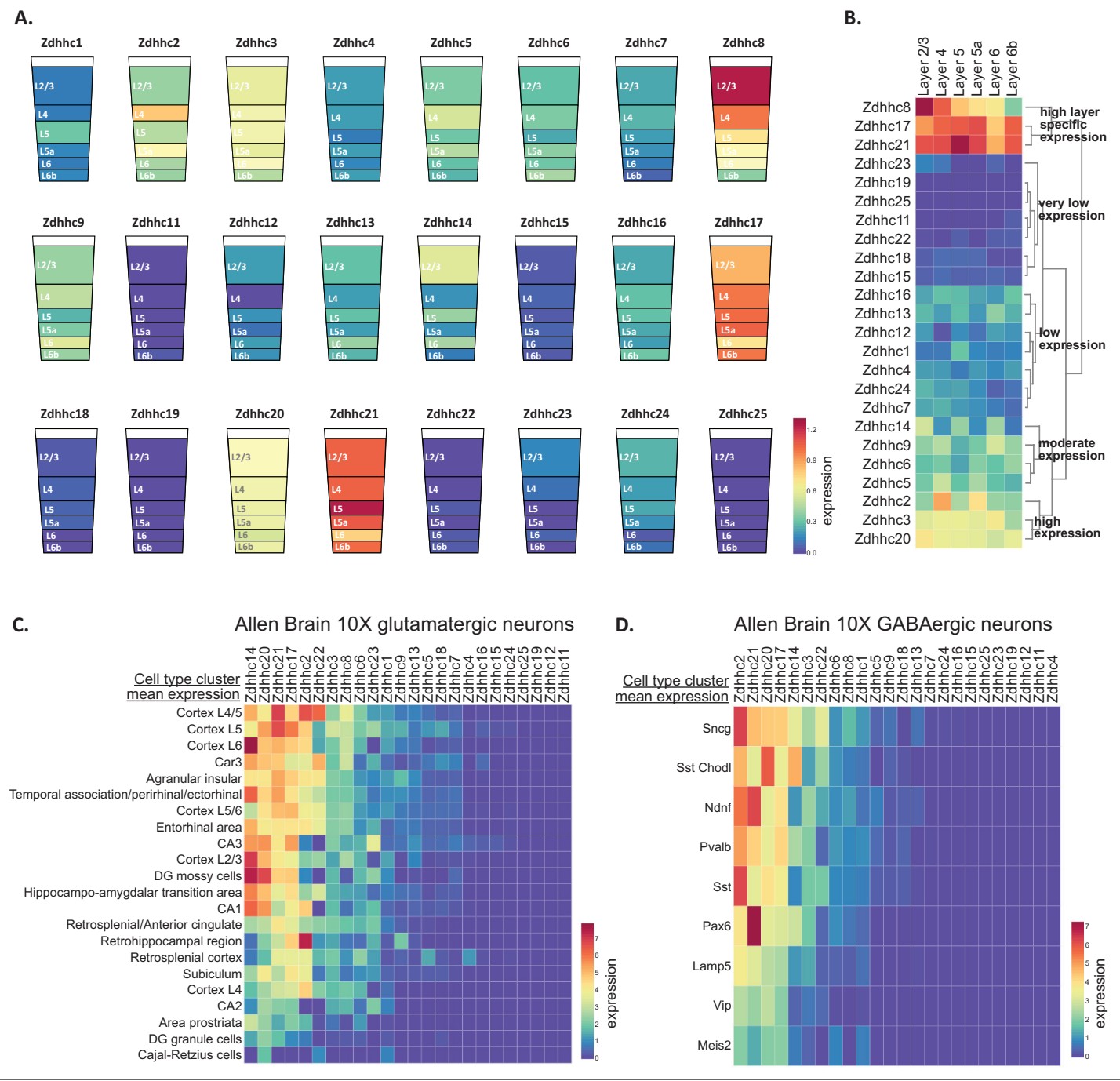

**Figure 3.** Pyramidal neuron layer specific ZDHHC expression. (**A**) Heatmap of excitatory neuron ZDHHC expression from somatosensory cortex (original data scRNAseq data from *Zeisel et al., 2015*) projected onto diagrams of cortical layers. (**B**) Hierarchical clustering of ZDHHC expression data in A. Heatmap units in (**A, B**): mean log2(counts per 10,000+1) (**C**) Heatmap of scRNAseq data from Allen Brain 10 X genomics (*Yao et al., 2021*). Data are represented as mean ZDHHC expression per excitatory neuron subtype, with columns and rows sorted by descending mean ZDHHC expression per row/column. (**D**) As in (**C**) but for inhibitory neuron subtypes. Heatmap units for (**C, D**): trimmed mean (25–75%) Log2(CPM +1).

The online version of this article includes the following source data and figure supplement(s) for figure 3:

**Source data 1.** Zdhhc expression cortical layers from '*Zeisel et al., 2015*' (Related to *Figure 3A and B*).

**Source data 2.** Zdhhc expression by excitatory neuron (subtype mean) from Allen Brain 10 X mouse dataset (Related to *Figure 3C*).

**Source data 3.** Zdhhc expression by inhibitory neuron (subtype mean) from Allen Brain 10 X mouse dataset (Related to *Figure 3D*).

**Figure supplement 1.** Heterogeneous ZDHHC expression in excitatory neurons of the cortex.

*Figure 3 continued*

**Figure supplement 1—source data 1.** Expression of Zdhhcs in cortical excitatory neurons across RNAseq studies (related to *Figure 3—figure supplement 1A*).

**Figure supplement 1—source data 2.** In situ hybridization images used from Allan Brain Atlas (related to *Figure 3—figure supplement 1B*).

ABHD17B, and ABHD17C). Compared with the ZDHHC enzymes, relatively less is known about the substrates, subcellular localization and brain expression patterns of this family of enzymes. We next explored BrainPalmSeq to determine which cell-types/brain regions show the highest expression of de-palmitoylating enzymes.

We first examined expression heatmaps for the known de-palmitoylating enzymes across the 265 cell-types identified in the 'MouseBrain' dataset (*Figure 4A*) and the cell-type averages for the 565 cell-types from the 'DropViz' dataset (*Figure 4—figure supplement 1A*). *Abhd12*, which is a membrane-bound enzyme that also has an important role in lysophosphatidylserine metabolism, and is associated with neurodegenerative disorder PHARC (polyneuropathy, hearing loss, ataxia, retinosis pigmentosa, and cataract; *Blankman et al., 2013*), had the broadest expression across all cell-types. This was followed by *Ppt1,* with expression of both of these enzymes being notably elevated in neurons of the hindbrain and immune cells. *Abhd4* was highly enriched in glial cells, particularly astrocytes, with low expression across neuronal subtypes in both datasets, consistent with previous reports of colocalization of *Abhd4* mRNA with astrocyte marker, *Slc1a3* (*László et al., 2020*). Ranked expression (sorted by descending averages) of de-palmitoylating enzymes in the 'MouseBrain' (*Figure 4B*) and 'DropViz' datasets (*Figure 4—figure supplement 1B*) revealed that *Abhd12*, *Abhd17a*, *Ppt1*, *Abhd17b*, and *Abhd16a* were the highest expressed depalmitoylating enzymes across studies. *Lypla2* expression was greater than *Lypla1* overall in the brain, with *Lypla2* expression being highest in neurons and ependymal cells (*Figure 4—figure supplement 1A*). *Abhd17c* had the lowest brain expression of all the de-palmitoylating enzymes studied. Correlation analysis between the ZDHHCs and de-palmitoylating enzymes revealed numerous instances of co-expression with almost every ZDHHC (*Figure 4D*), revealing potential cooperative pairs of palmitoylating and de-palmitoylating enzymes in the nervous system.

Although the ZDHHC enzymes are thought to act autonomously, several accessory proteins have been discovered that can regulate ZDHHC stability, localization and catalytic activity (*Salaun et al., 2020*). These include GOLGA7 (GCP16), which can bind to ZDHHC9 and enhance both protein stability and enzymatic activity by stabilizing the ZDHHC9 auto-palmitoylated intermediate that is formed prior to palmitate transfer from the enzyme to the substrate protein (*Mitchell et al., 2014*; *Swarthout et al., 2005*). Both GOLGA7 and related isoform GOLGA7B are also able to interact with ZDHHC5, with the latter influencing ZDHHC5 plasma membrane localization (*Woodley and Collins, 2019*). Finally, SELENOK (SELK; Selenoprotein K) is an ER localized protein that was found to interact with ZDHHC6, stabilize the auto-palmitoylated intermediate and increase palmitoylation of substrate proteins including the IP$_3$ receptor (*Fredericks et al., 2017*; *Fredericks et al., 2014*). We observed widespread expression of *Selenok* across all cell-types, with expression being considerably higher than any of the ZDHHCs, de-palmitoylating enzymes or other accessory proteins (*Figure 4A*, *Figure 4—figure supplement 1A*). This is consistent with the known functions of SELENOK in the ER-associated protein degradation pathway and regulation of ER calcium flux (*Pitts and Hoffmann, 2018*). *Golga7b* expression was widespread across neuronal subtypes but barely detected in glial cells (*Figure 4A*, *Figure 4—figure supplement 1A*). Accordingly, *Golga7b* expression was also strongly correlated with several of the ZDHHCs that were most highly expressed in neurons, including *Zdhhc3*, *Zdhhc8*, *Zdhhc17*, and *Zdhhc21* (*Figure 4D*). In contrast, *Golga7* was enriched in glial cells, particularly in oligodendrocytes, similar to ZDHHC9 for which GOLGA7 is a key accessory protein (*Figure 4A*, *Figure 4—figure supplement 1A*). Overall, the co-expression between putative Zdhhc/accessory protein pairs detailed above (*Zdhhc5/Golga7b*; *Zdhhc9/Golga7*; *Zdhhc6/Selenok*) was relatively weak (*Figure 4E*), therefore we plotted the expression values for each of the known pairs across 265 identified cell types from 'MouseBrain' dataset (*Figure 4—figure supplement 2*). We found that while *Zdhhc5* is expressed in several cell types with little-to-no expression of *Golga7b*, *Zdhhc9* and *Zdhhc6* were not expressed in the absence of their accessory proteins (*Golga7* and *Selenok*, respectively) in any cell type. Co-expression of *Zdhhc5/Golga7b* was highest in neuronal subtypes (*Figure 4—figure*

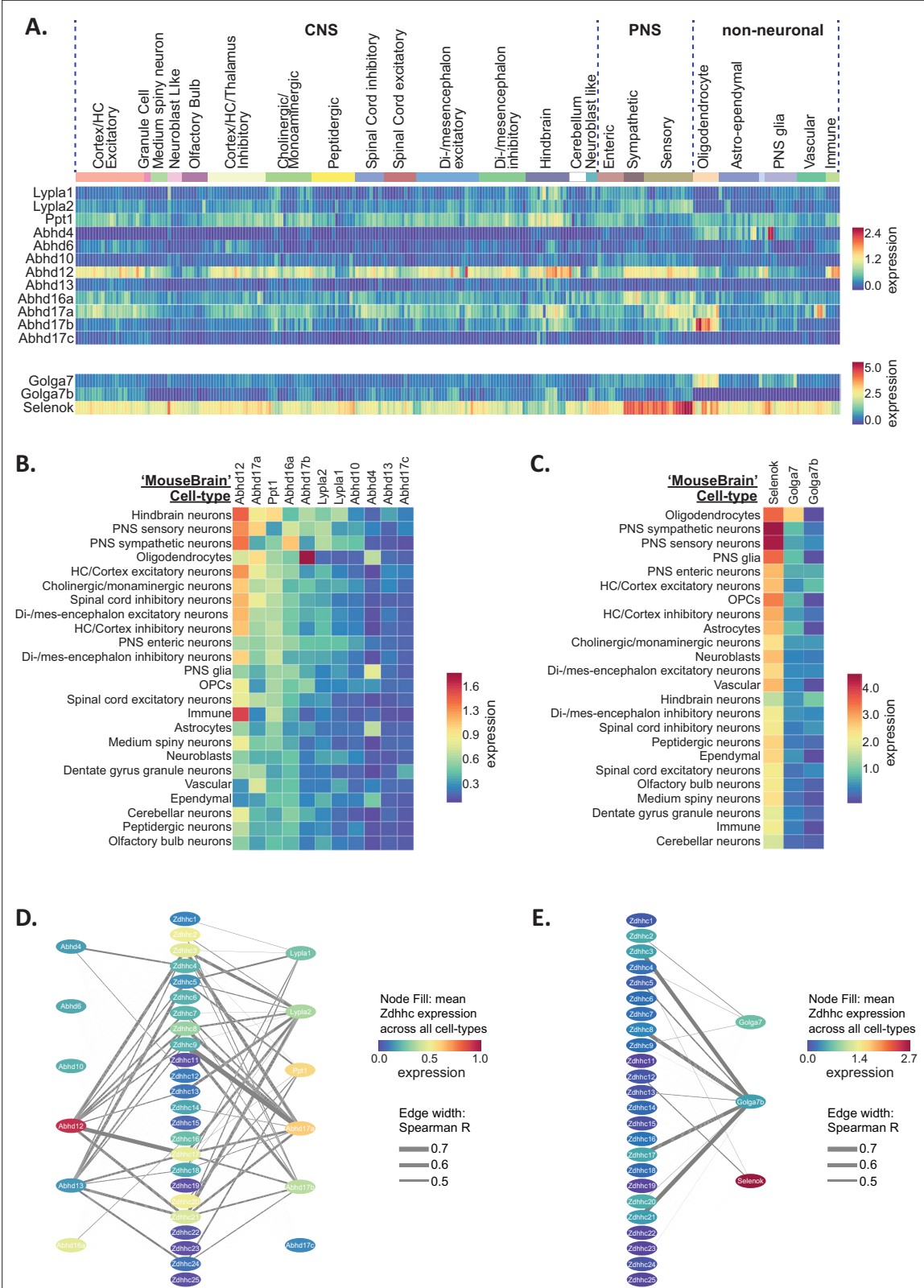

**Figure 4.** Heterogeneous de-palmitoylating enzyme and ZDHHC accessory protein expression in the mouse nervous system. (**A**) Heatmap showing expression of de-palmitoylating enzymes (top) and ZDHHC accessory subunits (bottom), extracted from scRNAseq study of mouse CNS and PNS (*Zeisel et al., 2018*). Each column represents one of the 265 metacells classified in the study. Metacells are organized according to hierarchical clustering designations generated by Zeisel et al. (**B**) Heatmap showing mean de-palmitoylating enzyme expression per hierarchical cluster, with columns and rows

*Figure 4 continued on next page*

*Figure 4 continued*

sorted by descending mean ZDHHC expression per row/column. (**C**) As B but for ZDHHC accessory proteins. (**D**) Correlation network showing ZDHHC co-expression with de-palmitoylating enzymes and accessory proteins across all metacells in 'MouseBrain' (Spearman *R*>0.4). Node color represents mean expression across all metacells. Edge thickness represents strength of correlation.

The online version of this article includes the following source data and figure supplement(s) for figure 4:

**Source data 1.** Depalmitoylating enzyme and Zdhhc accessory protein expression in the mouse nervous system from 'MouseBrain' dataset (related to *Figure 4A*).

**Source data 2.** Depalmitoylating enzyme and Zdhhc accessory protein expression (cell type averages) from 'MouseBrain' dataset (related to *Figure 4B and C*).

**Source data 3.** Spearman correlations of all palmitoylation associated genes from 'MouseBrain' dataset (related to *Figure 4D and E*).

**Figure supplement 1.** Heterogeneous de-palmitoylating enzyme and accessory protein expression in mouse brain.

**Figure supplement 1—source data 1.** De-palmitoylating enzyme and Zdhhc accessory expression (cell type mean) DropViz (related to *Figure 4—figure supplement 1*).

**Figure supplement 2.** Correlation of Zdhhc enzymes and their known accessory proteins.

**Figure supplement 3.** Cell-type enrichment of ZDHHC and depalmitoylating enzyme protein.

**Figure supplement 3—source data 1.** Calculation of cell type z-scores from *Rosenberg et al., 2018* RNAseq and *Sharma et al., 2015* proteomics (related to *Figure 4—figure supplement 3*).

*supplement 2A*), while *Zdhhc9/Golga7* co-expression was highest in oligodendrocytes, indicating there may be certain cellular contexts in which these pairings might be particularly important.

## Cell type enrichment of mRNA is partially maintained at the protein level

Previous studies have reported that mRNA and protein quantities can be poorly correlated due to post-transcriptional processing. These changes in protein degradation or synthesis can alter steady-state protein abundance (*Vogel and Marcotte, 2012*). We next set out to determine the extent to which the cell-type RNA expression patterns of the palmitoylation associated genes are maintained at the protein level, utilizing data from an available proteomic study of pooled isolated cells (MACS microbead plus antibody isolation) from neonatal mouse brains (*Sharma et al., 2015*), which we compared with the protein abundance with scRNAseq expression from similarly aged neonatal mice (*Rosenberg et al., 2018*). Focusing on proteins from our list of genes that were detected in all four cell types in the proteomic dataset, we calculated cell type z-scores for mRNA or protein abundance in each study and compared enrichment patterns (*Figure 4—figure supplement 3*). While we found numerous instances of similar patterns of cell-type enrichment for both mRNA and protein, including proteins that were primarily enriched in astrocytes (*Figure 4—figure supplement 3B*), microglia (*Figure 4—figure supplement 3C*), neurons (*Figure 4—figure supplement 3D*) and oligodendrocytes (*Figure 4—figure supplement 3E*), we also observed instances where this was not the case (*Figure 4—figure supplement 3F*). Notably, several highly expressed genes from the RNAseq study were not detected at the protein level in any cell type, including ZDHHC2, ZDHHC9 and SELENOK, which may be due to poor depth of coverage in the proteomic screen, or may indicate that these genes may be subject to more extensive post-transcriptional regulation. Overall, we found a significant correlation (*r*=0.63, p<0.0001) when plotting mRNA vs protein z-score across all cell types for the identified genes (*Figure 4—figure supplement 3G*). Together, these results indicate that the cell-type mRNA enrichment for many of the family of proteins that mediate *S*-palmitoylation is maintained at the protein level. Future advances in single-cell proteomics, along with tandem studies of both RNA and protein in the brain, will enable more precise comparison between the cell-type transcriptome and proteome.

## Loss-of-function mutations in palmitoylating and de-palmitoylating enzymes

Impaired regulation of *S*-palmitoylation has been implicated in numerous neurological disorders, many of which are due to loss-of-function (LOF) mutations in the genes encoding palmitoylating and de-palmitoylating enzymes (*Cho and Park, 2016*; *Matt et al., 2019*). We next sought to determine

if the regional and cell-type expression data available in BrainPalmSeq could reveal insights into the pathogenesis of disorders caused by LOF mutations in palmitoylating and de-palmitoylating enzymes. As many of these diseases have a neurodevelopmental origin, we examined whole brain datasets curated in BrainPalmSeq from the neonatal (*Rosenberg et al., 2018*), adolescent (*Zeisel et al., 2018*), and adult (*Sjöstedt et al., 2020*) mouse brain.

A single-nucleotide polymorphism (SNP) in the *ZDHHC8* gene has been implicated in increased susceptibility to schizophrenia (*Chen et al., 2004*; *Mukai et al., 2004*), while hemizygous microdeletion in the chromosomal locus 22q11, which encodes a number of genes including *ZDHHC8*, is one of the highest known genetic risk factors to developing schizophrenia (*Karayiorgou et al., 2010*). To assess the developmental expression of *Zdhhc8*, we averaged expression within broadly defined cell-type clusters that could be applied to both the Rosenberg and Zeisel scRNAseq datasets (*Figure 5A and B*; *Supplementary file 1*). *Zdhhc8* expression was highest in neurons of the cortex and hippocampus, followed by neurons of the mid- and hindbrain at both developmental ages. To explore regional expression in the adult mouse brain, we projected BrainPalmSeq generated heatmaps expression data from the 'Protein Atlas' mouse whole brain dataset (bulk RNAseq from major brain regions; *Figure 5—figure supplement 1*) onto anatomical maps of the mouse brain, again revealing highest expression of *Zdhhc8* in the cortex, followed by the hippocampus and basal ganglia (*Figure 5C*; *Sjöstedt et al., 2020*). *Zdhhc8* expression was particularly enriched in Layer 2/3 of the neonatal (not shown) and adult mouse (*Figure 3—figure supplement 1A*) cortex, which is the cortical layer with the most pronounced morphological deficits in patients with Schizophrenia (*Glantz and Lewis, 2000*; *Kolluri et al., 2005*; *Wagstyl et al., 2016*). Together, we found *Zdhhc8* expression patterns in the mouse brain that are established early in postnatal development and maintained into adulthood, that also overlay with many brain regions and cell types that are known to be severely affected in patients with schizophrenia. These observations support a model in which LOF *ZDHHC8* mutations may elicit many of the symptoms of schizophrenia by disrupting *S*-palmitoylation and normal neuronal development in these brain regions.

Mutations in the *ZDHHC9* gene, which is located on the X chromosome, have been identified in ~2% of individuals with X-linked intellectual disability (ID) (*Raymond et al., 2007*; *Tzschach et al., 2015*). Neuroanatomical abnormalities reported in patients with *ZDHHC9* mutations include decreased cortical, thalamic and striatal volume, as well as widespread white matter abnormalities with prominent hypoplasia (under-development) of the corpus callosum (*Baker et al., 2015*; *Bathelt et al., 2016*). Disrupted white matter integrity is thought to underlie deficits in global and local brain connectivity in patients with *ZDHHC9* mutations (*Bathelt et al., 2017*). *Zdhhc9* knock-out mice also develop similar pathological changes, including decreased volume of the corpus callosum (*Kouskou et al., 2018*). We observed considerable cell-type enrichment of *Zdhhc9* in oligodendrocytes across studies and developmental ages (*Figure 5D and E*), accompanied by moderate neuronal expression of *Zdhhc9* relative to other ZDHHCs across several brain regions including the hippocampus and cortex (*Figures 2A, 3A, 5D and E*), consistent with the known function of ZDHHC9 in regulating neuronal development (*Shimell et al., 2019*). Regionally, we found *Zdhhc9* expression in adult mice to be highly enriched in the corpus callosum, the largest white matter tract in the brain (*Figure 5F*). As myelin production by oligodendrocytes is critical for maintaining white matter integrity, these observations indicate that disrupting *S*-palmitoylation in oligodendrocytes may underlie the white matter pathology and decreased connectivity observed in patients with X-linked ID and *ZDHHC9* mutations.

Infantile neuronal ceroid lipofuscinosis (INCL or CLN1 disease) is a severe neurological disorder caused by LOF mutations in the *PPT1* gene that presents in the first 6–12 months of life and is characterized by rapid developmental regression, blindness and seizures, with continual deterioration until death in early childhood (*Nita et al., 2016*). While PPT1 is thought to primarily localize to lysosomes with an essential role in lysosomal degradation of *S*-palmitoylated proteins (*Lu et al., 1996*), this protein also has a synapse-specific function in regulating synaptic vesicle cycling and synaptic transmission (*Koster et al., 2021*; *Koster and Yoshii, 2019*). We found that neuronal *Ppt1* expression was high in postnatal neurons of the spinal cord, olfactory bulb, and mid/hindbrain, while microglia were the highest expressing non-neuronal cell type at both postnatal ages (*Figure 5G and H*). Neurodegeneration has been detected in the spinal cord prior to onset within the brain in *Ppt1* knock-out mice, accompanied by extensive glial cell activation including microgliosis, which is a pathological hallmark of CLN1 disease (*Shyng et al., 2017*). Mid-/hindbrain neurons also had high expression of

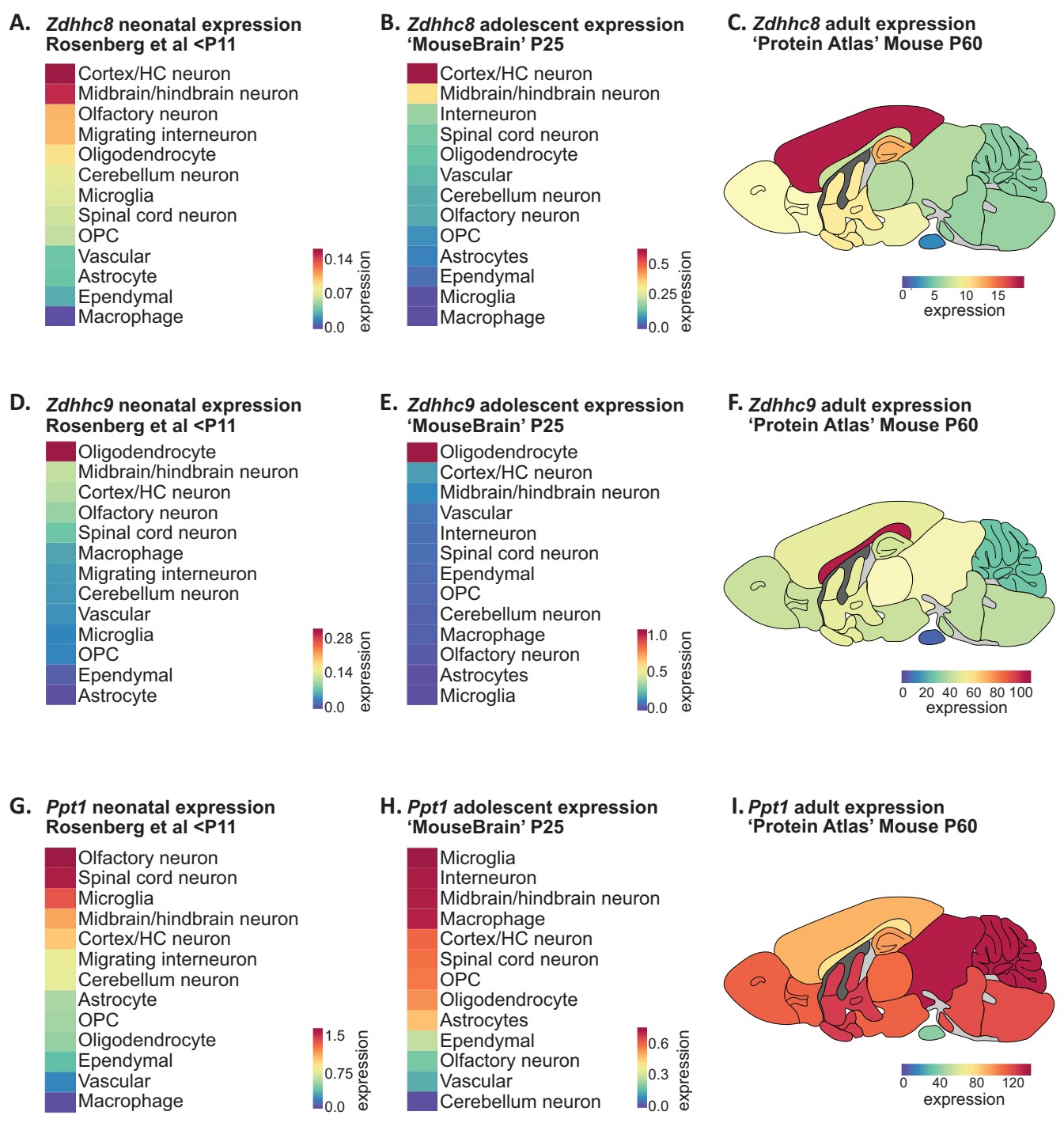

**Figure 5.** Disease associated palmitoylating enzyme regional and cell-type expression overlays with brain pathology in associated LOF disorders. (**A**) Heatmap showing ranked *Zdhhc8* expressing neuronal and glial cell types in descending order. Original data from scRNAseq neonatal mouse brain study; *Rosenberg et al., 2018*. Cell types were averaged as described in *Supplementary file 1*. Heatmap units: mean log2(counts per 10,000+1). (**B**) As in A, but original data from scRNAseq adolescent mouse brain study *Zeisel et al., 2018*. Heatmap units: mean log2(counts per 10,000+1). (**C**) Heatmap of *Zdhhc8* expression from whole brain regional bulk RNAseq data (original data from 'Protein Atlas'; *Sjöstedt et al., 2020*) projected onto anatomical map of mouse brain. Heatmap units: pTPM. (**D–F**) As in (**A**)-(**C**) but for *Zdhhc9*. (**G–I**) As in (**A**)-(**C**) but for *Ppt1*.

The online version of this article includes the following source data and figure supplement(s) for figure 5:

**Source data 1.** Disease associated palmitoylating enzyme regional and cell-type expression patterns (related to *Figure 5*).

**Figure supplement 1.** Anatomical sampling for bulk RNAseq study.

*Ppt1*, consistent with reports that *Ppt1* knock-out mice show early signs of brain pathology in the thalamus (*Kielar et al., 2007*). Overall, we observed widespread *Ppt1* expression in almost every brain region in adult mice, consistent with the sweeping neurological deficits associated with CLN1 disease (*Figure 5I*). Together, these observations reveal how the loss of *Ppt1* in cell types with high *Ppt1* expression may lead to cell death/dysfunction in the early stages of CLN1 disease.

## ZDHHC cell type enrichments can be used to predict and validate ZDHHC substrates

We next tested if ZDHHC expression patterns identified from BrainPalmSeq could be used to predict and validate *S*-palmitoylation substrates for regionally enriched ZDHHCs. We focused on *Zdhhc9*, which showed a consistent cell-type enrichment in oligodendrocytes across multiple studies in Brain-PalmSeq, while LOF mutations in *ZDHHC9* are known to result in reduced white matter integrity in the brain (*Raymond et al., 2007*). Examination of the Marques et al oligodendrocyte-specific scRNAseq dataset curated in BrainPalmSeq revealed that oligodendrocyte *Zdhhc9* expression increased throughout maturation, with highest expression in the myelin forming (MFOL) intermediate-maturity subtype of oligodendrocytes, while slightly lower expression is maintained in mature oligodendrocytes (MOL; *Figure 6A*; *Marques et al., 2016*).

To identify potential substrates for ZDHHC9, we cross-referenced a list of MFOL/MOL enriched genes identified in the study by *Marques et al., 2016* with the SwissPalm database to identify known palmitoylation substrates in these cell types (Swiss Palm Annotated; *Figure 6B*; *Blanc et al., 2019*; *Blanc et al., 2015*). PANTHER analysis of cellular component enrichments for these substrates revealed the most significant enrichment was for the term 'myelin sheath' (30 proteins; *Figure 6C*, *Figure 6—figure supplement 1*). To determine if any of the myelin sheath associated proteins could be palmitoylated by ZDHHC9, we selected three proteins (MOBP, PLP1 and CNP) for experimental validation (*Figure 6C*). We separately expressed each of these candidate substrates together with ZDHHC9 and its accessory protein GOLGA7 in HEK293T cells, and determined the proportion of palmitoylated substrate using an acyl-resin assisted capture (acyl-RAC) palmitoylation assay (*Forrester et al., 2011*). Co-expression of HA-ZDHHC9 and FLAG-GOLGA7 increased the palmitoylated fraction of MOBP and PLP1, indicating that these proteins are substrates for ZDHHC9 (*Figure 6D, E and G*). Conversely, CNP was not identified as a ZDHHC9 substrate in our assay (*Figure 6F and G*). While ZDHHC9 is able to palmitoylate MOBP and PLP1 in HEK cells, we observed no change in the palmitoylation of these substrates in *Zdhhc9* knock-out mice (P23 half brain) compared to wild-type (*Figure 6—figure supplement 2*). It is possible that ZDHHC9 is not required for the palmitoylation of these proteins in vivo (i.e. compensation by other ZDHHC enzymes), or that the detection of differences in protein palmitoylation requires isolation of specific cell populations that can be diluted using whole-brain lysate as has been previously reported (*Gorenberg et al., 2022*; *Kouskou et al., 2018*). Overall, these results demonstrate how the cell-type enrichments of ZDHHC enzymes identified in this study can be used, along with the lists of similarly enriched palmitoylation substrates, to guide the identification of enzyme-substrate interactions that can be further investigated in vivo.

## Discussion

### BrainPalmSeq as a tool to generate hypotheses about proteins that control S-palmitoylation in the brain

We have demonstrated the utility of BrainPalmSeq by providing examples of how this database can be used to explore detailed region and cell-type-specific expression patterns of the known palmitoylating and de-palmitoylating enzymes, and their accessory proteins. We reveal how these expression patterns can be used to predict/validate *S*-palmitoylation substrates and better understand diseases associated with loss of function mutations in the enzymes that mediate *S*-palmitoylation. Given the number of brain regions and cell types incorporated into BrainPalmSeq that were not discussed in the present study, including the thalamus, hypothalamus, amygdala, striatum, and cerebellum, there is rich potential for users to explore the data and generate hypotheses about the role of these enzymes in the brain.

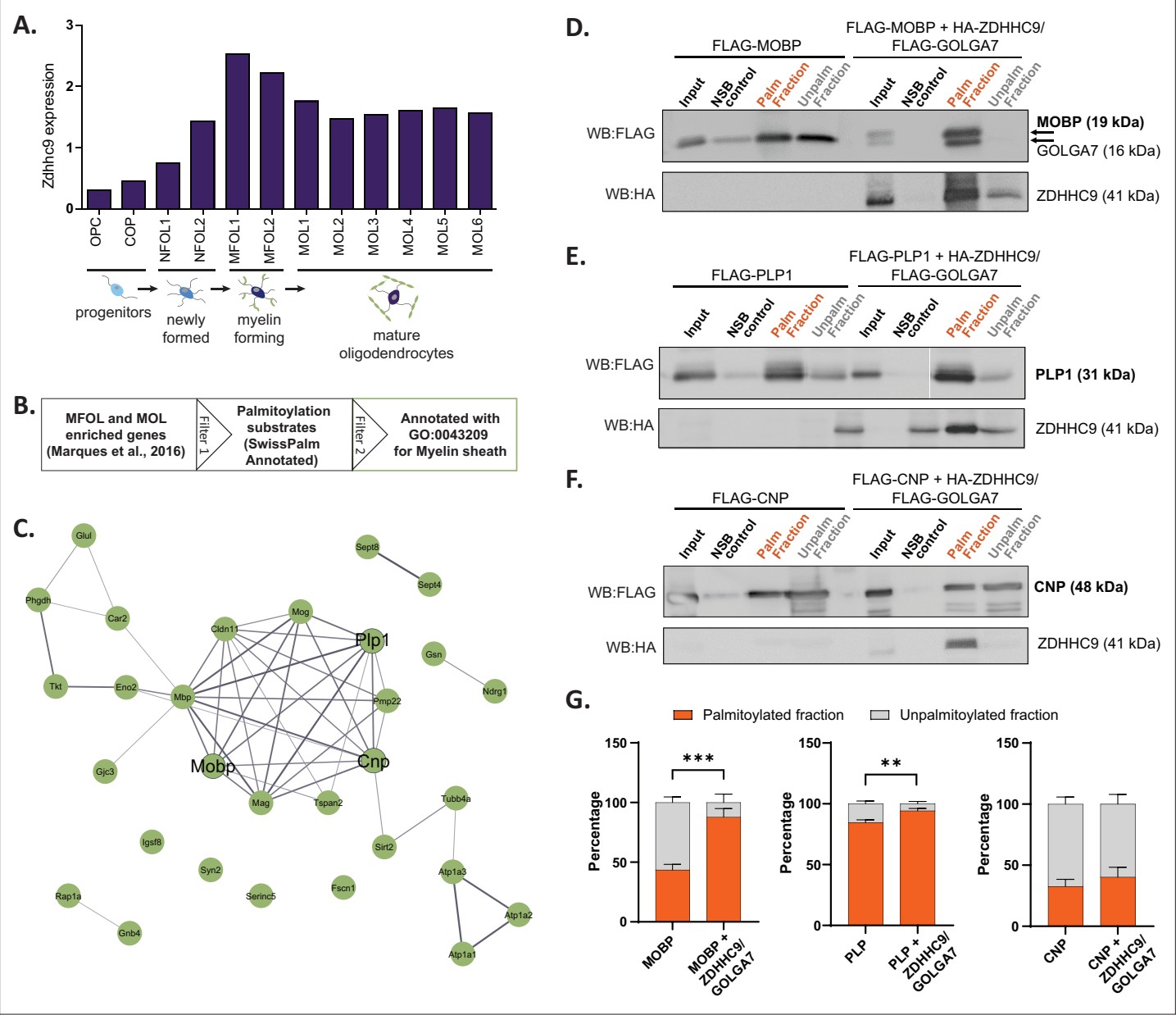

**Figure 6.** Validation of projected *S*-palmitoylation substrates of *Zdhhc9* derived from cell-type enriched expression. (**A**) Graph of expression data for *Zdhhc9* extracted from BrainPalmSeq. Original data from oligodendrocyte scRNAseq study by Marques et al. Expression units: mean log2(counts per 10,000+1). (**B**) Diagram illustrating workflow to generate a list of oligodendrocyte enriched palmitoylation substrates, GO annotated for myelin sheath for experimental validation. (**C**) STRING diagram of myelin sheath annotated palmitoylation substrates. (**D**) Western blot following Acyl-RAC palmitoylation assay in HEK293 cells to identify palmitoylated and unpalmitoylated fractions of FLAG-MOBP either without or with co-transfection of FLAG-GOLGA7 and HA-ZDHHC9. Input = unprocessed protein lysate. NSB control = non-specific binding of protein to sepharose resin in control pipeline. Palm fraction = palmitoylated protein. Unpalm fraction = unpalmitoylated protein. (**E–F**) As in (**D**) but for FLAG-PLP1 (**E**) or FLAG-CNP (**F**). (**G**) Graphs quantifying the ratio of palmitoylated to unpalmitoylated protein either with or without co-transfections with FLAG-GOLGA7 and HA-ZDHHC9. n=4–6 HEK cell cultures per condition. Statistics shown for palmitoylated fraction. Two-way ANOVA; Šídák's *post hoc*; mean ± SEM. MOBP: p=0.0004, 95% CI −0.6594 to −0.2266; PLP1: p=0.0046, 95% CI −0.1660 to −0.03011; CNP: p=0.6981, 95% CI −0.3274–0.1742.

The online version of this article includes the following source data and figure supplement(s) for figure 6:

**Source data 1.** Zdhhc9 expression in oligodendrocytes, *Marques et al., 2016*.

**Source data 2.** *S*-palmitoylation substrates enriched in myelinating and mature oligodendrocytes (related to ***Figure 6B and C***).

**Source data 3.** Acyl Rac quantification for Zdhhc9 substrate validation (related to ***Figure 6G***).

**Source data 4.** Uncropped raw and annotated western blot images for all data quantified in ***Figure 6G***.

*Figure 6 continued on next page*

*Figure 6 continued*

**Figure supplement 1.** GO cellular component analysis for oligodendrocyte enriched genes.

**Figure supplement 2.** Palmitoylation of myelin proteins in *Zdhhc9* KO mice.

**Figure supplement 2—source data 1.** Acyl-Rac quantification/western blots of mouse brain lysates from wild-type and Zdhhc9 KO mice (related to *Figure 6—figure supplement 2*).

## Insights into the role of *S*-palmitoylation associated enzymes in brain physiology and pathology

While we found that many of the proteins we studied showed correlated expression across the entire mouse nervous system, particularly those enriched in neurons including *Zdhhc3*, *Zdhhc8*, *Zdhhc17*, and *Zdhhc21*, expression of these genes was segregated within more narrowly defined neuronal populations such as the excitatory pyramidal neurons within the hippocampal tri-synaptic loop or layers of the somatosensory cortex. This is in line with the extensive neuronal transcriptional heterogeneity identified recently by a number of scRNAseq studies (*Saunders et al., 2018*; *Yao et al., 2021*; *Zeisel et al., 2018*; *Zeisel et al., 2015*). The genes that determine neuronal identity fall under four broad functional categories: those that control transcriptional programs, membrane conductance, neurotransmission, or synaptic connectivity (*Zeisel et al., 2018*). We also report heterogeneity in the neuronal fingerprint of palmitoylating and de-palmitoylating enzyme expression, which will in turn give rise to differential *S*-palmitoylation of neuronal proteins. Future work is needed to determine how these specific ZDHHC expression patterns are related to dynamic *S*-palmitoylation in these neuronal sub-types, and how the elevated expression of certain ZDHHCs can alter neuronal function. Given that *S*-palmitoylation is a key regulator of neuronal development, and that nearly half of all known synaptic proteins are substrates for palmitoylation (*Sanders et al., 2015*), this heterogeneity is likely to be a key mechanism in the fine tuning of neuronal function and synaptic transmission.

Many of the ZDHHCs that we observed with consistently elevated expression across multiple studies in BrainPalmSeq have already been studied in the context of neuronal signaling, including ZDHHC2, ZDHHC3, ZDHHC8, and ZDHHC17 (*Ji and Skup, 2021*; *Matt et al., 2019*). In contrast, ZDHHC20 and ZDHHC21 are relatively understudied in the nervous system, despite our observation that these are two of the most abundantly expressed ZDHHCs across neuronal cell types, with broad expression of *Zdhhc20* also in glial cells. A recent study defined a role for ZDHHC21 in the palmitoylation of serotonergic receptor 5-HT1A and implicated downregulation of ZDHHC21 in the development of major depressive disorder (*Gorinski et al., 2019*). Interestingly, both ZDHHC20 and ZDHHC21 have a potential role in the pathogenesis of Alzheimer's disease, as they can palmitoylate BACE1, Tau and amyloid precursor protein (*Cho and Park, 2016*). Further work is required to understand the likely important role of these enzymes in the brain.

We made several other interesting observations during our examination of BrainPalmSeq that were not discussed in detail in the present study but we believe warrant further investigation. For example, the particularly elevated expression of *Zdhhc2* in peripheral sensory neurons may indicate an important role for palmitoylation in this cell type. Across multiple studies we observed striking enrichment of *Zdhhc14* in cerebellar Purkinje neurons, a cell type in which *S*-palmitoylation is known to be important for long-term depression, although the role of ZDHHC14 in this process has not yet been investigated (*Thomas et al., 2013*). *Zdhhc23* was similarly enriched in the CA2 region of the hippocampus, with comparatively low expression across other cell types. More broadly, the elevated expression of a variety of palmitoylating enzymes in central neurons that utilize acetylcholine or monoamines as neurotransmitters would suggest an important role for *S*-palmitoylation in these neurons that has yet to be explored. Accordingly, many of the key proteins involved in cholinergic synaptic transmission are *S*-palmitoylation substrates including muscarinic acetylcholine receptor M2 (CHRM2), acetylcholinesterase (ACHE), and ATP-citrate synthase (ACLY; *Blanc et al., 2015*; *Blanc et al., 2019*). Our observations of co-enrichment of certain palmitoylating and de-palmitoylating enzymes are also of interest, such as *Abhd17b* and *Zdhhc9* in oligodendrocytes. It is possible that these enzymes share substrates to mediate dynamic palmitoylation/de-palmitoylation, or conversely, have separate substrates in order to maintain stable *S*-palmitoylation states of certain oligodendrocyte expressed proteins. Importantly, the data accessibility in BrainPalmSeq will enable researchers to develop hypotheses regarding their cell type, brain region or protein of interest.

The palmitoylome of each cell type in the nervous system is likely to be highly heterogeneous and will be determined by the expression of both the *S*-palmitoylation substrates and the palmitoylating and de-palmitoylating enzymes in a given cell type. Furthermore, accumulating evidence has revealed that this palmitoylome can be altered by extrinsic factors such as chronic stress and neuronal activity (*Kang et al., 2008*; *Zareba-Koziol et al., 2019*). While we have provided projected palmitoylomes composed of several highly expressed or enriched *S*-palmitoylation substrates in select brain regions and cell types, experimental validation to reveal the relative palmitoylation of substrates under various conditions is needed to fully understand these cellular differences. Nevertheless, we were able utilize our projected palmitoylomes to validate substrates for ZDHHC9, providing insight into the potential role of this enzyme in myelin regulation in the brain.

Neurological disorders that arise from LOF gene mutations may be predicted to lead to pathological changes that are more severe in the brain regions in which these genes are most highly expressed. We observed this type of regional overlay for the expression patterns of *Zdhhc8*, *Zdhhc9,* and *Ppt1*. Numerous other brain disorders are thought to be exacerbated by an imbalance in *S*-palmitoylation, such as decreased *S*-palmitoylation of HTT in Huntington's disease (*Virlogeux et al., 2021*; *Yanai et al., 2006*), increased *S*-palmitoylation of APP and TAU in Alzheimer's disease (*Cho and Park, 2016*), and reduced *S*-palmitoylation of 5-HTA receptor in major depressive disorder (*Gorinski et al., 2019*). Efforts are already underway to normalize aberrant *S*-palmitoylation in neurological diseases in order to improve clinical outcomes (*Roberts et al., 2012*; *Virlogeux et al., 2021*). Understanding the brain expression patterns of the enzymes that mediate palmitoylation in these diseases will be paramount to developing and targeting such therapeutics.

While scRNAseq has proven to be a groundbreaking tool that has helped identify many unique cell types in the brain while allowing a means to study cellular gene expression patterns at remarkable depth, characterization of the proteome of individual brain cells has lagged behind transcriptome profiling. This is largely due to the inability to amplify protein in the same way as RNA, greatly reducing the depth of coverage observed compared with transcriptomic analyses. Instead, proteomic studies generally utilize pooled cell isolation methods to improve detection (*Goto-Silva and Junqueira, 2021*; *Wilson and Nairn, 2018*). Proteomic analysis is particularly important as previous studies have found that RNA and protein abundance are often moderately correlated, as a result of post-translational processing of protein or RNA molecules. Our comparison of mRNA and protein abundance in the major brain cell types during postnatal development revealed numerous instances of similar cell-type enrichments for the majority of the proteins detected. This would imply that while the absolute amount of mRNA and protein may differ in a given cell type, the mRNA cell-type enrichments are predictive of protein enrichment for many of the genes included in our study.

## Differential gene expression as a means to control *S*-palmitoylation in the brain

The mechanisms that govern ZDHHC enzyme-substrate interactions are complex and still not fully understood. While the majority of post-translational modifications including phosphorylation and N-glycosylation are highly sequence specific (*Schwarz and Aebi, 2011*; *Ubersax and Ferrell, 2007*), several studies have revealed that *S*-palmitoylation by ZDHHCs can be stochastic, proximity based and lacking in stereo-selectivity (*Rocks et al., 2010*; *Rodenburg et al., 2017*). Contrasting studies have shown that numerous ZDHHCs have specific protein interacting domains including ankyrin repeat (AR), PDZ and SH3 domains that facilitate substrate interactions, providing support for a model in which more specific enzyme-substrate interactions can govern *S*-palmitoylation (*Abrami et al., 2017*; *Lemonidis et al., 2015*; *Plain et al., 2020*; *Rana et al., 2019*; *Thomas et al., 2012*; *Verardi et al., 2017*). Furthermore, a recent study found striking substrate specificity for several ZDHHCs with the G-protein subunit Gαo, and revealed intriguing observations that the subcellular localization of a number of *S*-palmitoylation substrates could be controlled by changing the localization, and importantly, the expression of certain ZDHHC enzymes. In this study, *S*-palmitoylated substrates were found to accumulate in the subcellular compartment in which their partner ZDHHCs were targeted (*Solis et al., 2022*). This is particularly relevant as the ZDHHCs are known to have diverse subcellular localizations including the Golgi, ER, endosomes and plasma membrane (*Globa and Bamji, 2017*). Transcriptional control of differentially compartmentalized palmitoylating and de-palmitoylating enzymes could therefore be an essential mechanism for regulating the subcellular localization, and function,

of *S*-palmitoylated protein substrates. Accordingly, LOF mutations in certain ZDHHC enzymes leads to cell-type-specific disruption in *S*-palmitoylation that is not compensated by other members of the large ZDHHC family. We provide a means to investigate the expression of the proteins that regulate *S*-palmitoylation, making BrainPalmSeq an invaluable resource to both researchers and clinicians that are working to better understand the role of *S*-palmitoylation in the brain.

# Materials and methods

**Key resources table**

| Reagent type (species) or resource | Designation | Source or reference | Identifiers | Additional information |
|---|---|---|---|---|
| Recombinant DNA reagent | FLAG-GOLGA7 | Gift from Maurine Linder Washington University School of Medicine | | |
| Recombinant DNA reagent | HA-ZDHHC9 | *Shimell et al., 2019* | | |
| Recombinant DNA reagent | FLAG-MOBP | Origene, Maryland, USA | CAT#: RC223946 | |
| Recombinant DNA reagent | FLAG-PLP1 | Origene, Maryland, USA | CAT#: RC218616 | |
| Recombinant DNA reagent | FLAG-CNP | Origene, Maryland, USA | CAT#: RC207038 | |
| Cell line (*Homo-sapiens*) | HEK293T | ATCC | CAT # CRL-1573 | |
| Antibody | Anti-HA, (rabbit monoclonal) | Cell Signaling Technology | CAT#: 3,724 | Dilution 1:1000 |
| Antibody | Anti-FLAG (mouse monoclonal) | Origene | CAT#: TA50011-100 | Dilution 1:1000 |
| Antibody | Anti-MOBP (Rabbit polyclonal) | Invitrogen | CAT# PA5-100618 | Dilution 1:1000 |
| Antibody | Anti-PLP1 (Rabbit polyclonal) | Abcam | CAT# ab28486 | Dilution 1:1000 |
| Antibody | Anti-CNP (Mouse monoclonal) | Abcam | CAT# ab6319 | Dilution 1:1000 |
| Commercial assay or kit | CAPTUREome S-palmitoylated protein kit | Badrilla, UK | CAT# K010-311 | |

## Data processing for BrainPalmSeq

For *Zeisel et al., 2018* ('MouseBrain'), single-cell counts (UMI from 3' end counting) were downloaded from MouseBrain.org (loom file named l5_all.loom), and log normalized by first scaling the expression values provided to a sum of 10,000 per cell before calculating log2(scaled_counts +1). Averages were then performed by brain region, neurotransmitter and taxonomy for each gene.

For *Saunders et al., 2018* ('DropViz), Metacell counts were downloaded from DropViz.org (count file metacells.BrainCellAtlas_Saunders_version_2018.04.01.RDS and annotation file annotation.Brain-CellAtlas_Saunders_version_2018.04.01.RDS) and log normalized by first scaling the expression values provided to a sum of 10,000 per metacell before calculating log2(scaled_counts +1). Averages were then performed by cell type, tissue and class for each gene. Genes associated with palmitoylation were selected in order to create the heatmaps.

For *Zeisel et al., 2015*, single-cell counts (UMI from 3' end counting) were downloaded from https://storage.googleapis.com/linnarsson-lab-www-blobs/blobs/cortex/expression_mRNA_17-Aug-2014.txt, and log normalized by first scaling the expression values provided to a sum of 10,000 per cell before calculating log2(scaled_counts +1). Averages were then performed by cluster, tissue and class for each gene. Genes associated with palmitoylation were selected in order to create the heatmaps, categories comprising fewer than 5 single cells are not displayed.

For *Marques et al., 2016*, single-cell counts (UMI from 3' end counting) were downloaded from GEO with accession ID GSE75330 (file GSE75330_Marques_et_al_mol_counts2.tab) and log normalized by first scaling the expression values provided to a sum of 10,000 per cell before calculating log2(scaled_counts +1). Averages were then performed by cluster and region for each gene. Genes associated with palmitoylation were selected in order to create the heatmaps, categories comprising fewer than 5 single cells are not displayed.

For *Rosenberg et al., 2018*, single-cell counts (UMI from 3' end counting) were downloaded from GSE110823, and log normalized by first scaling the expression values provided to a sum of 10,000

per cell before calculating log2(scaled_counts +1). Averages were then performed by brain region, neurotransmitter and taxonomy for each gene. Genes associated with palmitoylation were selected in order to create the heatmaps.

For *Sjöstedt et al., 2020*, expression data were downloaded as Protein-coding transcripts per million (pTPM) from proteinatlas.org ('RNA mouse brain region gene data') and not further processed.

For *Cembrowski et al., 2016* ('Hipposeq'), expression data were downloaded as FPKM directly from hipposeq.janelia.org and were not further processed.

For Allen Brain 10X data, expression data were downloaded as trimmed means (25–75%) Log2(CPM +1) from https://portal.brain-map.org/atlases-and-data/rnaseq/mouse-whole-cortex-and-hippocampus-10x and were not further processed.

## Correlation analysis

Spearman correlation values between genes and their significances were calculated in R using the expression results obtained for each cell type as described above.

## Identification of *S*-palmitoylation substrates with SwissPalm

Gene lists were inputted into SwissPalm (https://swisspalm.org/proteins) input file function and cross-referenced with 'Dataset 3: Palmitoylation validated or found in at least one palmitoyl-proteome (SwissPalm annotated)' for *Mus musculus*, with an additional filter for UniProt 'Reviewed' proteins.

## Generating a projected palmitoylome for dorsal hippocampus

To curate the regionally enriched projected-palmitoylome, the enrichment analysis tools in hipposeq (https://hipposeq.janelia.org/) were used to compare each of the selected Cell Lines vs the other Cell Lines in the analysis (Selected Cell Lines = dorsal DG, CA3, CA2 and CA1), with the following parameters: 'Fold threshold'=1.5; 'FPKMmin threshold'=5, 'FDR'=0.05. The resulting lists of regionally enriched transcripts were cross referenced with SwissPalm as described above to identify regionally enriched *S*-palmitoylation substrates.

## Bioinformatic analysis

Gene Ontology (GO) analysis was performed using statistical overrepresentation tests in PANTHER16.0 (*Mi et al., 2009*) with default settings and *Mus musculus* as the reference species. Biological process GO terms were extracted and ranked according to false discovery rate (FDR). Kyoto Encyclopedia of Genes and Genomes (KEGG) analysis was performed using the web-based program Enrichr (*Chen et al., 2013*; *Kuleshov et al., 2016*) and ranked according to -log Adjusted P-value. Synaptic Gene Ontologies (SynGO; version 1.1) analysis was performed using default settings with brain expressed genes as a background and terms for 'biological process' were ranked according to -log Adjusted p-value. Functional protein interaction networks were identified using the Search Tool for the Retrieval of Interacting Genes (STRING) 11.0 (*Szklarczyk et al., 2019*) with *Mus musculus* as the reference species. Seven types of protein interactions were used for network generation, including text mining, neighborhood, co-occurrence, co-expression, gene fusion, experiments, and databases.

## Data presentation

Heatmaps within the manuscript were plotted and hierarchical clustering performed in Displayr (https://www.displayr.com) using the 'Dendrogram' function. Cytoscape (Version 3.8.0) was used to draw correlation networks.

## Heatmap creation for BrainPalmSeq

All plots for the BrainPalmSeq database were generated using curated RNA sequencing datasets. Python 3 and Javascript scripts were used with the plotting library Bokeh to generate the interactive heatmaps to display and compare these datasets on the BrainPalmSeq website (*Bokeh Development Team, 2018*).

## Cell culture

HEK293T cells were purchased from ATCC (CRL-1573) and the cell line was authenticated by STR profiling and confirmed negative for mycoplasma. Cells were thawed and aliquoted into a 10 cm dish

with 10 mL prewarmed (37 °C) DMEM (GIBCO, Thermo Fisher Scientific, Waltham, MA) supplemented with 10% fetal bovine serum (FBS) (GIBCO, Thermo Fisher Scientific, Waltham, MA) and 1% Pen/Strep (P/S) (GIBCO, Thermo Fisher Scientific, Waltham, MA). HEK293T cells were then placed in a 37 °C incubator with 5% $CO_2$ and passaged approximately every 5 days, or once confluency was achieved.

### Transfection

A total of 70–80% confluent HEK293T cells were transfected using Lipofectamine 2000 (Invitrogen/Life Technologies, Carlsbad, CA) according to the manufacturer's recommendations. Each well of a six-well plate was transfected with a total of 3 µg DNA, 150 µL of Opti-Mem (GIBCO, Thermo Fisher Scientific, Waltham, MA) was used with 6 µL of Lipofectamine 2000 (Invitrogen/Life Technologies, Carlsbad, CA). Experimental condition wells were transfected with 1 µg of the indicated construct of interest, 1 µg of HA-ZDHHC9 (mouse; *Shimell et al., 2019*), and 1 µg of FLAG-GOLGA7 (Maurine Linder, Washington University School of Medicine). Human FLAG-MOBP (CAT#: RC223946), FLAG-PLP1 (CAT#: RC218616) and FLAG-CNP (CAT#: RC207038) were acquired from Origene, Maryland, USA. Control condition wells were transfected with 1 µg of the indicated construct of interest, and 2 µg of a scrambled control plasmid. Cells were lysed using the acyl-RAC assay lysis buffer 48 hr after transfection.

### Palmitoylation assay (acyl-RAC)

The commercially available CAPTUREome S-palmitoylated protein kit (Badrilla, Leeds, UK) was used in accordance with the manufacturer's guidelines with three optimizations: (1) prior to the cell lysis step, wells were washed with 1 mL of 1 X PBS to eliminate any dead cells or residual media; (2) during the cell lysis step, DNase (Sigma-Aldrich, St. Louis, MO), was added to the solution (5 µL per 500 µL of lysis buffer); and (2) protein concentration was measured prior to the separation of experimental sample and negative control sample using the BCA Assay (Pierce, Thermo Fisher Scientific, Waltham, MA).

### Western blot analysis

Western blotting was performed using 4% stacking and 12% resolving SDS-PAGE gels. PVDF membranes were then blocked for 1 hr at room temperature with 5% BSA in 0.05% TBS-T. PVDF membranes were incubated with the indicated primary antibodies (anti-HA: Cell Signaling Technology, C29F4, Rabbit mAb CAT#: 3724, 1:1000; anti-FLAG: Origene, mouse monoclonal antibody, CAT#: TA50011-100, 1:1000) overnight at 4 °C. Proteins were then visualized using enhanced chemiluminescence (Immubilon Western Chemiluminescent HRP Substrate) on a BioRad ChemiDoc XRS +scanner. Blots were then quantified using Fiji1 software. The palmitoylated and unpalmitoylated fractions were calculated using the following equations respectively: 100*(Palm Fraction – NSB/ ((Palm Fraction – NSB)+Unpalm Fraction)) and 100*(Unpalm Fraction / (Palm Fraction +Unpalm Fraction)). Criteria for data inclusion were sufficient transfection/antibody signal and minimal protein in the non-specific binding control.

### Animals

All experimental procedures and housing conditions were approved by the UBC Animal Care Committee and were in accordance with the Canadian Council on Animal Care (CCAC) guidelines. Male C57BL/6 J mice (Jackson Laboratory, Sacramento, CA) were bred with female heterozygous *Zdhhc9* knockout mice (B6;129S5-Zdhhc9tm1Lex/Mmucd) obtained from the Mutant Mouse Resource and Research Center (MMRRC) at the University of California, Davis. Mice were bred and genotyped as described in *Shimell et al., 2019*. For acyl-RAC assays, litter matched knockout and wild-type males were used at P22-P24.

### Preparation of brain lysate

Brains were removed and immediately flash frozen in isopentane (–55 °C). Half brains (not including cerebellum) were lysed in the acyl-RAC kit blocking buffer first using a pestle and then passing through a 26-gauge needle. The samples were then processed following the acyl-RAC protocol as described above.

### Statistical analysis

Spearman correlations were performed with the function cor in R, and their significances were obtained using the function cor.test followed by a correction for multiple testing using p.adjust with

the fdr method (Benjamini & Hochberg). For assessing enrichment of *S*-palmitoylation substrates within Zdhhc co-expressed genes, Fisher's exact test was performed using the function fisher.test in R.

For validation of substrates using Acyl-Rac palmitoylation assay, two-way ANOVA with Šídák's post hoc was used to assess significance of both palmitoylated and unpalmitoylated fractions. Statistics are shown for palmitoylated fraction. No outliers were excluded. Statistical analyses were performed in GraphPad Prism 9.2.0 (San Diego, CA, USA). *n* represents the number of individual HEK cell culture dishes. Each culture dish is defined as a biological replicate. No technical replicates were performed.

## Acknowledgements

The authors thank Drs. A Ciernia, M Cembrowski, T Murphy and J LeDue for helpful discussion. Funding This work was supported by CIHR Foundation grant (F18-00650) to SXB and by computational resources made available through the NeuroImaging and NeuroComputation Centre at the Djavad Mowafaghian Centre for Brain Health (RRID:SCR_019086) and the Dynamic Brain Circuits in Health and Disease Research Excellence Cluster DataBinge Forum.

## Additional information

### Funding

| Funder | Grant reference number | Author |
|---|---|---|
| Canadian Institutes of Health Research | FDN-148468 Foundation Grant | Peter W Hogg<br>Kurt Haas |
| Canadian Health Services Research Foundation | F18-00650 CIHR Foundation Grant | Angela R Wild<br>Glory G Nasseri<br>Rocio B Hollman<br>Danya Abazari<br>Shernaz X Bamji |

The funders had no role in study design, data collection and interpretation, or the decision to submit the work for publication.

### Author contributions

Angela R Wild, Conceptualization, Data curation, Formal analysis, Investigation, Visualization, Methodology, Writing - original draft; Peter W Hogg, Software, Visualization, Methodology; Stephane Flibotte, Data curation, Formal analysis, Visualization, Methodology; Glory G Nasseri, Rocio B Hollman, Formal analysis, Investigation; Danya Abazari, Investigation; Kurt Haas, Supervision, Writing - review and editing; Shernaz X Bamji, Conceptualization, Supervision, Funding acquisition, Project administration, Writing - review and editing

### Author ORCIDs

Angela R Wild http://orcid.org/0000-0002-2125-4443
Peter W Hogg http://orcid.org/0000-0003-2176-4977
Kurt Haas http://orcid.org/0000-0003-4754-1560
Shernaz X Bamji http://orcid.org/0000-0003-0102-9297

### Decision letter and Author response

Decision letter https://doi.org/10.7554/eLife.75804.sa1
Author response https://doi.org/10.7554/eLife.75804.sa2

## Additional files

### Supplementary files

• Transparent reporting form

• Supplementary file 1. Comparable category assignments for cell types identified in *Rosenberg et al., 2018*; *Zeisel et al., 2018*. Related to *Figure 5*.

## Data availability

All data generated or analysed during this study are included in the manuscript and supporting file; Source Data files have been provided for all Figures.

The following previously published datasets were used:

| Author(s) | Year | Dataset title | Dataset URL | Database and Identifier |
|---|---|---|---|---|
| Karolinska Institute | 2018 | Single cell sequencing of the whole adult mouse brain | https://trace.ncbi.nlm.nih.gov/Traces/index.html?view=study&acc=SRP135960 | Sequence Read Archive, SRP135960 |
| Saunders A, McCarroll S | 2018 | A Single-Cell Atlas of Cell Types, States, and Other Transcriptional Patterns from Nine Regions of the Adult Mouse Brain | https://www.ncbi.nlm.nih.gov/geo/query/acc.cgi?acc=GSE116470 | NCBI Gene Expression Omnibus, GSE116470 |
| Sjöstedt | 2020 | RNA mouse brain subregion sample gene data | https://www.proteinatlas.org/download/rna_mouse_brain_hpa.tsv.zip | The Human Protein Atlas, rna_mouse_brain_sample_hpa |
| Rosenberg A, Roco C | 2018 | Single-cell profiling of the developing mouse brain and spinal cord with split-pool barcoding | https://www.ncbi.nlm.nih.gov/geo/query/acc.cgi?acc=GSE110823 | NCBI Gene Expression Omnibus, GSE110823 |
| Cembrowski M, Spruston N | 2016 | Hipposeq: a comprehensive RNA-seq database of gene expression in hippocampal principal neurons | https://www.ncbi.nlm.nih.gov/geo/query/acc.cgi?acc=GSE74985 | NCBI Gene Expression Omnibus, GSE74985 |
| Zhang Y, Chen K, Sloan SA, Scholze AR, Caneda C, Ruderisch N, Deng S, Daneman R, Barres BA | 2014 | An RNA-sequencing transcriptome and splicing database of glia, neurons, and vascular cells of the cerebral cortex | https://www.ncbi.nlm.nih.gov/geo/query/acc.cgi?acc=GSE52564 | NCBI Gene Expression Omnibus, GSE52564 |
| Zeisel A, Muñoz Manchado AB, Lönnerberg P, Linnarsson S | 2015 | Cell types in the mouse cortex and hippocampus revealed by single-cell RNA-seq | https://www.ncbi.nlm.nih.gov/geo/query/acc.cgi?acc=GSE60361 | NCBI Gene Expression Omnibus, GSE60361 |
| Yao Z, van Velthoven CT, Nguyen TN, Goldy J, Sedeno-Cortes AE, Baftizadeh F, Bertagnolli D, Casper T, Chiang M, Crichton K, Ding S, Fong O, Garren E, Glandon A, Gouwens NW, Gray J, Graybuck LT, Hawrylycz MJ, Hirschstein D, Kroll M, Lathia K, Lee C, Levi B, McMillen D, Mok S, Pham T, Ren Q, Rimorin C, Shapovalova N, Sulc J, Sunkin SM, Tieu M, Torkelson A, Tung H, Ward K, Dee N, Smith KA, Tasic B, Zeng H | 2021 | A taxonomy of transcriptomic cell types across the isocortex and hippocampal formation | https://www.ncbi.nlm.nih.gov/geo/query/acc.cgi?acc=GSE185862 | NCBI Gene Expression Omnibus, GSE185862 |

*Continued on next page*

*Continued*

| Author(s) | Year | Dataset title | Dataset URL | Database and Identifier |
|---|---|---|---|---|
| Kozareva V, Martin C, Osorno T, Rudolph S, Guo C, Vanderburg C, Nadaf N, Regev A, Regehr W, Macosko E | 2021 | A transcriptomic atlas of mouse cerebellar cortex comprehensively defines cell types | https://www.ncbi.nlm.nih.gov/geo/query/acc.cgi?acc=GSE165371 | NCBI Gene Expression Omnibus, GSE165371 |
| Schulmann A, Phillips JW, Hara E, Wang L, Lemire AL, Nelson SB, Hantman AW | 2019 | A repeated molecular architecture across thalamic pathways | https://www.ncbi.nlm.nih.gov/geo/query/acc.cgi?acc=GSE133911 | NCBI Gene Expression Omnibus, GSE133911 |
| Chen R, Wu X, Jiang L, Zhang Y | 2017 | Single-cell RNA-seq reveals hypothalamic cell diversity | https://www.ncbi.nlm.nih.gov/geo/query/acc.cgi?acc=GSE87544 | NCBI Gene Expression Omnibus, GSE87544 |
| Stanley GM, Treutlein B, Quake SR | 2016 | Cellular Taxonomy of the Mouse Striatum as Revealed by Single-Cell RNA-Seq | https://www.ncbi.nlm.nih.gov/geo/query/acc.cgi?acc=GSE82187 | NCBI Gene Expression Omnibus, GSE82187 |
| Cembrowski M | 2020 | Extensive and spatially variable within-cell-type heterogeneity across the basolateral amygdala | https://www.ncbi.nlm.nih.gov/geo/query/acc.cgi?acc=GSE148866 | NCBI Gene Expression Omnibus, GSE148866 |
| Marques S, Zeisel A, Linnarsson S, Castelo-Branco G | 2016 | RNA-seq analysis of single cells of the oligodendrocyte lineage from nine distinct regions of the anterior-posterior and dorsal-ventral axis of the mouse juvenile central nervous system | https://www.ncbi.nlm.nih.gov/geo/query/acc.cgi?acc=GSE75330 | NCBI Gene Expression Omnibus, GSE75330 |
| Martirosyan A | 2020 | Identification of region-specific astrocyte subtypes at single cell resolution | https://www.ncbi.nlm.nih.gov/geo/query/acc.cgi?acc=GSE114000 | NCBI Gene Expression Omnibus, GSE114000 |
| Li Q, Barres B | 2019 | Deep single-cell RNAseq of microglia and brain myeloid cells from various brain regions and developmental stages | https://www.ncbi.nlm.nih.gov/geo/query/acc.cgi?acc=GSE123025 | NCBI Gene Expression Omnibus, GSE123025 |
| Schulmann A, Phillips JW, Hara E, Wang L, Lemire AL, Nelson SB, Hantman AW | 2019 | Single-cell atlas of thalamic projection systems | https://www.ncbi.nlm.nih.gov/geo/query/acc.cgi?acc=GSE133912 | NCBI Gene Expression Omnibus, GSE133912 |

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
