## [Editor Report]

This paper will be of broad interest to neuroscientists, providing a rich resource for future research. Using available RNAseq data the authors build an easy-to-work-with web platform which will enable researchers to survey the expression patterns of palmitoylating and de-palmitoylating enzymes and their potential co-expressed substrates within the mouse nervous system. Using this map, the authors test hypotheses about the relationship between these enzymes and neurological diseases and generate hypotheses about enzyme/substrate relationships based on expression correlations.

---

## [Decision Letter]

**Decision letter after peer review:**

Thank you for submitting your article "BrainPalmSeq: A curated RNA-seq database of palmitoylating and depalmitoylating enzyme expression in the mouse brain" for consideration by *eLife*. Your article has been reviewed by 3 peer reviewers, one of whom is a member of our Board of Reviewing Editors, and the evaluation has been overseen by Suzanne Pfeffer as the Senior Editor. The reviewers have opted to remain anonymous.

Essential revisions:

Experimental:

1. The result presented on Figure 6. should be validated by loss of function experiments.

2. At a few instances protein-level validation of RNA expression should strengthen the authors' conclusions.

3. It is unclear why certain established depalmitoylating enzymes were excluded from the database. At least some of them should be added.

Explanation:

4. Please outline the long-term strategy for the future management of the database, explain how it will be updated with newly available experimental data.

Modification of the text:

5. Additional details regarding the discussion are detailed in the body of Rev. #3's report.

The authors are also invited to address all other points raised by the reviewers in their rebuttal.

*Reviewer #1 (Recommendations for the authors):*

Could the authors comment on cross-validation with ISH? When is it possible, beneficial etc?

Do the authors plan to update the database with new datasets as they become available?

The authors could provide a clear description of the unique benefits of their web tool in comparison with other tools. What types of comparisons are available only with this tool? Ease of use? etc.

Only 2 types of hippocampal inhibitory interneurons are included. Is there any data available about additional inhibitory cell types or interneurons in general, i.e. cells identified with inhibitory specific cell markers (e.g. GAD65,67)?

*Reviewer #2 (Recommendations for the authors):*

– The main recommendation regards the possibility of the authors to greatly improve the robustness of their predictions by using available ZDHHC antibodies to test and validate the protein expression of certain ZDHHC enzymes in different neuro-associated tissue culture cell types.

*Reviewer #3 (Recommendations for the authors):*

1. No information is provided about the long-term integrity of the web resource 'brainpalmseq'. Is the corresponding author's institution committing to maintaining this in perpetuity?

2. I appreciate that the current analysis is a snapshot of known palmitoylating and depalmitoylating enzymes, but I don't see a justification to exclude ABHD10 (PMID: 31740833) from the analysis. Add this enzyme to Figure 4.

3. Line 307 (or discussion). It is important to comment on the very modest correlation of expression of Golga7b / Golga7 with either of the two zDHHC-PATs they have been suggested to regulate. This would appear to imply they are 'optional' rather than 'obligatory' partners, correct?

4. Figure 6. If both MOBP and PLP are zDHHC9 substrates how can you explain the relatively low palmitoylation of MOBP (~40% increased to ~90% by zDHHC9 & Golga7) but relatively high palmitoylation of PLP (~80% increased to ~90%) in the same cell type? Could other zDHHC-PATs be involved?

5. Figure 6. I appreciate that the increase in PLP palmitoylation is statistically significant, but is the very small change in palmitoylation likely to be biologically significant? Can you be confident that there are no non-specific effects of overexpressing zDHHC9? These experiments would be significantly improved by silencing zDHHC9 (or repeating with the zDHHC9 knockout animals that this lab used in PMID: 31747610).

---

## [Author Response]

Essential revisions:Experimental:1. The result presented on Figure 6. should be validated by loss of function experiments.

As per the reviewers’ suggestion, we have now done loss of function experiments to determine whether ZDHHC9 is not only *sufficient* to palmitoylate proteins that we have predicted to be ZDHHC9 substrates, but also *necessary*. Reviewer 2 was specifically concerned about unspecific S-acylation, (‘This setup is known to often provide unspecific S-acylation events which result from excess enzyme or substrate availability’*)* and Reviewer 3 recommended repeating the experiment using *Zdhhc9* knockout mice.

As suggested by Reviewer 3, we performed acyl-rac assays on hemi-brains (half brain minus cerebellum) from postnatal day 23 WT and *Zdhhc9* KO mice. However, using this method we observed no changes in the palmitoylation of the 3 substrates tested, PLP1, MOBP and CNP (see Figure 6—figure supplement 2). This can be attributed to many factors, including compensation by other ZDHHCs that are also expressed in oligodendrocytes (e.g. ZDHHC20), or the use of lysate from a variety of brain regions that may ‘dilute’ the observable effects that are occurring in specific brain regions or cell populations. Of note, the palmitoylation of Ras has also been reported to be unchanged in whole brains of Zdhhc9 knockout mice (PMID: 29944857), and Ras is a very well-established substrate for ZDHHC9 (e.g. PMID: 16000296; PMID: 24127608). Compensation has been attributed to a lack of phenotype in many knockout mouse lines and does not negate the fact that our acyl-rac assays demonstrate that ZDHHC9 *can* palmitoylate MOBP and PLP1. It is possible that acyl-rac assays done specifically using oligodendrocytes from WT and KO brains may uncover differences in substrate palmitoylation, however this experiment is particularly challenging due to the large amount of starting tissue required for the acyl-rac assay and is beyond the intended scope of the present study.

We understand Reviewer 2’s concern about data that relies on overexpression of substrates and ZDHHC enzymes. However, we respectfully submit that this type of experiment is well established and a good starting point to determine whether a particular ZDHHC can palmitoylate a substrate. As a counter to the idea that unspecific S-acylation events may result from excess enzyme or substrate availability, in our assay CNP was not palmitoylated when overexpressed in HEK cells together with ZDHHC9 (i.e. only 2 of the 3 substrates tested were palmitoylated by ZDHHC9). Moreover, we have previously overexpressed ZDHHCs together with another putative substrate (δ-catenin) and demonstrated that only 2 of the 19 ZDHHCs tested could palmitoylate that substrate (see Brigidi, et al., 2014, Figure 7). Similar experiments have been done to identify palmitoylating and depalmitoylating enzymes for PSD-95 (PMID: 15603741, PMID: 27307232).

Our study seeks to illustrate the utility of our web-tool and how users might use bioinformatics to begin investigating potential enzyme-substrate interactions. We believe we have shown this in Figure 6 and do not see a need to add this additional experiment that is subject to the caveats outlined above. However, we would be happy to include this negative data in a supplemental figure if the editors and reviewers deem it necessary.

2. At a few instances protein-level validation of RNA expression should strengthen the authors' conclusions.

We agree with reviewers that mRNA and protein levels are not always well correlated and that instances of protein-level validation would be an excellent addition to the study. We were unable to validate protein expression using immunohistochemistry as available antibodies for the ZDHHC family are not of sufficient quality to discern the expression patterns with acceptable specificity to compare directly with extremely high resolution RNAseq data. While characterization of the proteome of individual brain cells has lagged behind transcriptome profiling due to the inability to amplify proteins in the same way as RNA, in an attempt to address this point, we turned to a proteomic study that identified the proteome of pooled neonatal brain cells isolated by MACS microbead/antibody cell separation (PMID: 26523646). We compared the cell type enrichments of the detected palmitoylation associated proteins included in our study with cell-type averages of RNA expression from an age matched scRNAseq study (PMID: 29545511). Overall, we found a significant correlation between the cell type enrichments of both RNA and protein in the brain for palmitoylation associated proteins detected in the proteomic screen. We believe that the use of open proteomic data from isolated cell types is preferable to the use of low specificity antibodies when comparing protein enrichment patterns with RNAseq data. We have included this as a supplementary figure (Figure 4—figure supplement 3). A new Results section has also been added to line 364 and discussion to line 563.

3. It is unclear why certain established depalmitoylating enzymes were excluded from the database. At least some of them should be added.

We agree with the reviewers that an extended list of de-palmitoylating enzymes should be included in both the manuscript and website. To this end, we have updated the manuscript and figures to include the depalmitoylating enzymes that are blocked by Palmostatin B and hexadecylfluorophosphonate (HDFP; PMID: 28630138 ), which includes all of the enzymes suggested by the reviewers (ABHD4, ABHD6 ABHD10, ABHD12) with the exception of ABHD11, which was weakly inhibited by these blockers and for which we unable to find a significant body of evidence supporting the depalmitoylating activity of this enzyme (although we will reconsider if the reviewers feel strongly about including this enzyme). We have however still included ABHD11 in the website heatmaps and data download for future investigation of those interested. Also included in the website and manuscript (although not requested by the reviewers) are ABHD13 and ABHD16A, which are both inhibited by HDFP and have been shown recently to depalmitoylate SNAP25 (unpublished data, Jennifer Greaves lab). In the event that future genes are discovered, we will update the website according to such feedback (see strategy below).

Explanation:4. Please outline the long-term strategy for the future management of the database, explain how it will be updated with newly available experimental data.

The website will be maintained and updated for a minimal period of 5 years by first author Research Associate, Angela Wild. This will include adding new genes to the website, particularly newly identified accessory proteins and de-palmitoylating enzymes, as they are discovered/validated. While we feel that the current studies included in the website provide sufficient coverage of the whole brain at a variety of developmental ages, we will add new datasets to the website that offer substantial improvements to the data curated. For example, we will consider studies that are broader in their scope offering new insights about brain cell development or regional heterogeneity, or those that feature improved techniques that considerably increase resolution of gene expression detection offering novel information. To facilitate updates, we have added a ‘Contact Us’ feature to the website which will allow users to suggest new genes or datasets to add. In the event of A. Wild’s absence, a replacement student/staff member will be fully trained to perform these duties. The fully automated heatmap code used in the website (created by author Peter Hogg, PhD student at UBC) to display the interactive heatmaps can be easily updated by uploading a new gene list into the code. The collaboration to create the heatmaps for the website was initiated/managed by Jeff LeDue, who is head of the Managing Director of the Neuroimaging and Neurocomputation Centre at UBC and coordinator of the UBC Dynamic Brain Circuits in Health and Disease Research Excellence Cluster. This collaboration will continue if additional resources are needed in the future management of the website. We will continue to collaborate with Stephane Flibotte who is a permanent staff member at the UBC LSI Bioinformatics Facility, or with his colleagues, to process new datasets that we acquire according to the analysis pipeline utilized throughout.

Modification of the text:5. Additional details regarding the discussion are detailed in the body of Rev. #3's report.

We thank Rev. #3 for their extremely detailed review of our manuscript and feel that their discussion points have greatly helped to clarify many interpretations of the data. Each point is addressed in detail below.

The authors are also invited to address all other points raised by the reviewers in their rebuttal.Reviewer #1 (Recommendations for the authors):Could the authors comment on cross-validation with ISH? When is it possible, beneficial etc?

We agree that ISH comparisons are useful for cross validation of RNA expression patterns and had previously included several examples that compare Zdhhc RNAseq expression data with ISH from the Allen brain https://mouse.brain-map.org/, in Figure 2—figure supplement 1 (hippocampus) and Figure 3—figure supplement 1 (somatosensory cortex). We observed many instances of regionally restricted mRNA expression in the RNAseq datasets curated that was reflected in the Allen Brain ISH, supporting the use of RNAseq to generate cellular maps of gene expression.

Do the authors plan to update the database with new datasets as they become available?

We do plan to update the database with new datasets as they become available. See major discussion point 4 above.

The authors could provide a clear description of the unique benefits of their web tool in comparison with other tools. What types of comparisons are available only with this tool? Ease of use? etc.

In the manuscript we have already outlined the barriers and difficulties with previously available tools (page 4 line 95): “We found however, there were several barriers to the easy access for much of this data, with no single resource available to evaluate multi-study expression data. Data can also be difficult to access when studies are not accompanied by an interactive online web viewer, while datasets that do have a web viewer employ diverse interfaces that are often complex, particularly for large scRNAseq datasets. Furthermore, the differing study specific analysis pipelines, as well as the variety of data presentation formats in web viewers including heatmaps, bar charts, tables or t-SNE plots can make datasets difficult for non-bioinformaticians to interpret and compare.” And have further outlined how our tool improves accessibility: “In order to remove these barriers and provide easy access to expression data for the proteins that regulate S-palmitoylation in the brain, we created ‘BrainPalmSeq’, an easy-to-use web platform allowing user driven interrogation of compiled multi-study expression data at cellular resolution through simple interactive heatmaps that are populated according to user selected brain regions, cell types or genes of interest (http://brainpalmseq.med.ubc.ca/).”

Only 2 types of hippocampal inhibitory interneurons are included. Is there any data available about additional inhibitory cell types or interneurons in general, i.e. cells identified with inhibitory specific cell markers (e.g. GAD65,67)?

We believe there may be a misunderstanding on this point. Included in our study is the detailed scRNAseq dataset of neurons performed by the Allen Institute, which identified 122 GABAergic neuron subtypes in the cortex and hippocampus. These data are summarized for the Zdhhc enzymes in Figure 3D of the manuscript and the full genelist/dataset are available within the webtool (https://brainpalmseq.med.ubc.ca/brain-regions/neocortex-allen-brain-atlas-rnaseq/search-allen-brain-map-by-gabaergic-neuron-subtypes/).

Reviewer #2 (Recommendations for the authors):– The main recommendation regards the possibility of the authors to greatly improve the robustness of their predictions by using available ZDHHC antibodies to test and validate the protein expression of certain ZDHHC enzymes in different neuro-associated tissue culture cell types.

We thank the reviewer for this suggestion as this suggestion has strengthened the manuscript. Please see major discussion point 2 above for a complete response.

Reviewer #3 (Recommendations for the authors):1. No information is provided about the long-term integrity of the web resource 'brainpalmseq'. Is the corresponding author's institution committing to maintaining this in perpetuity?

We have now added our plan for long-term maintenance of this resource (see Essential revision point 4).

2. I appreciate that the current analysis is a snapshot of known palmitoylating and depalmitoylating enzymes, but I don't see a justification to exclude ABHD10 (PMID: 31740833) from the analysis. Add this enzyme to Figure 4.

The reviewer makes an excellent point and we appreciate this suggestion. We have addressed this point more completely in Essential revision point 3.

3. Line 307 (or discussion). It is important to comment on the very modest correlation of expression of Golga7b / Golga7 with either of the two zDHHC-PATs they have been suggested to regulate. This would appear to imply they are 'optional' rather than 'obligatory' partners, correct?

We are hesitant to over-interpret the correlation patterns to draw any firm conclusions about whether an accessory protein is optional or obligatory on the basis of bioinformatic correlation analysis alone. However, as this will be of interest to many groups working in palmitoylation, we have performed additional analysis on the relationships between ZDHHCs and known accessory proteins. We have added a new supplementary figure (Figure 4—figure supplement 2) that plots the expression values for each of the known ZDHHC/accessory protein pairs across 265 identified cell types from ‘MouseBrain’ dataset. These graphs reveal that while *Zdhhc5* is expressed in several cell types with little-to-no expression of *Golga7b*, both *Zdhhc9* and *Zdhhc6* were always co-expressed with their accessory proteins (*Golga7* and *Selenok*, respectively). This analysis also reveals the cell types in the brain with the highest co-expression between Zdhhcs and accessory partners, raising the possibility that there are certain cellular contexts in which pairing might be more important for regulating ZDHHC function. We have updated the results (line 354) to reflect these findings.

4. Figure 6. If both MOBP and PLP are zDHHC9 substrates how can you explain the relatively low palmitoylation of MOBP (~40% increased to ~90% by zDHHC9 & Golga7) but relatively high palmitoylation of PLP (~80% increased to ~90%) in the same cell type? Could other zDHHC-PATs be involved?

The reviewer is correct that there are differences in basal palmitoylation of MOBP, PLP and CNP in our HEK cell assay is variable. Numerous PATs are known to be expressed in HEK293T cells (https://sites.broadinstitute.org/ccle/datasets) and PLP may be highly palmitoylated basally by endogenous ZDHHC9 or ZDHHC9 plus another PAT(s). The level of endogenous palmitoylation does not necessarily speak to multiple PATs, but merely the extent to which a substrate is palmitoylated.

5. Figure 6. I appreciate that the increase in PLP palmitoylation is statistically significant, but is the very small change in palmitoylation likely to be biologically significant? Can you be confident that there are no non-specific effects of overexpressing zDHHC9? These experiments would be significantly improved by silencing zDHHC9 (or repeating with the zDHHC9 knockout animals that this lab used in PMID: 31747610).

It is very difficult to determine biological impact based on the magnitude of change. As per reviewers’ request, we have repeated our work with *Zdhhc9* knockout mice (See Essential revision point 1).